# Nuclear translocation of p85β promotes tumorigenesis of *PIK3CA* helical domain mutant cancer

Yujun Hao [1,2,3,6✉], Baoyu He[1,6], Liping Wu[2,3,4,6], Yamu Li[2,3,6], Chao Wang[2,3], Ting Wang[1], Longci Sun[1], Yanhua Zhang[1], Yangyang Zhan [1], Yiqing Zhao[2,3], Sanford Markowitz[3,5], Martina Veigl[3], Ronald A. Conlon[2,3] & Zhenghe Wang [2,3✉]

PI3Ks consist of p110 catalytic subunits and p85 regulatory subunits. *PIK3CA*, encoding p110α, is frequently mutated in human cancers. Most *PIK3CA* mutations are clustered in the helical domain or the kinase domain. Here, we report that p85β disassociates from p110α helical domain mutant protein and translocates into the nucleus through a nuclear localization sequence (NLS). Nuclear p85β recruits deubiquitinase USP7 to stabilize EZH1 and EZH2 and enhances H3K27 trimethylation. Knockout of p85β or p85β NLS mutant reduces the growth of tumors harboring a *PIK3CA* helical domain mutation. Our studies illuminate a novel mechanism by which *PIK3CA* helical domain mutations exert their oncogenic function. Finally, a combination of Alpelisib, a p110α-specific inhibitor, and an EZH inhibitor, Tazemetostat, induces regression of xenograft tumors harboring a *PIK3CA* helical domain mutation, but not tumors with either a WT *PIK3CA* or a *PIK3CA* kinase domain mutation, suggesting that the drug combination could be an effective therapeutic approach for *PIK3CA* helical domain mutant tumors.

[1] State Key Laboratory of Oncogenes and Related Genes, Shanghai Cancer Institute, Renji Hospital, Shanghai Jiao Tong University School of Medicine, 200032 Shanghai, China. [2] Department of Genetics and Genome Sciences, Case Western Reserve University, 10900 Euclid Avenue, Cleveland, OH 44106, USA. [3] Case Comprehensive Cancer Center, School of Medicine, Case Western Reserve University, 10900 Euclid Avenue, Cleveland, OH 44106, USA. [4] Department of Chemistry, College of Basic Medical Sciences, Army Medical University (Third Military Medical University), Chongqing 400038, P.R. China. [5] Department of Medicine, School of Medicine, Case Western Reserve University, 10900 Euclid Avenue, Cleveland, OH 44106, USA. [6] These authors contributed equally: Yujun Hao, Baoyu He, Liping Wu, Yamu Li. ✉email: yjhao@shsci.org; zxw22@case.edu

*P*IK3CA, which encodes the p110α catalytic subunit of PI3 kinase, is one of the most frequently mutated oncogenes in human cancers[1–3]. Recently, the FDA approved the combination of p110α-specific inhibitor Alpelisib and the estrogen receptor antagonist Fulvestrant to treat *PIK3CA*-mutant breast cancer patients whose tumors are hormone receptor (HR)-positive and HER2-negative[4]. This approval highlights mutant *PIK3CA* (encoding p110α protein) as a critical cancer drug target and the importance of further understanding the molecular mechanisms by which *PIK3CA* mutations drive tumorigenesis. PI3Ks are heterodimers consisting of a p110 catalytic subunit and a p85 regulatory subunit[5]. Normally, the p85 subunits bind and stabilize the p110 subunit and inhibit its enzymatic activity[6]. Upon growth factor stimulation, the SH2 domains of p85 bind to the phospho-tyrosine residues on receptor protein kinases or adaptor proteins such as insulin receptor substrate 1 (IRS1), which activates PI3K and catalyzes the conversion of phosphatidylinositol-4,5-bisphosphate (PIP2) to phosphatidylinositol-3,4,5-triphosphate (PIP3)[5]. PIP3 recruits pleckstrin homology domain-containing proteins, including PDK1 and AKTs, to the cell membrane to activate signaling pathways[5].

Most tumor-derived *PIK3CA* mutations are clustered in two hotspots in p110α protein: the E542, E545, and Q546 residues in the helical domain and the H1047 residue in the kinase domain[1,7]. Accumulating evidence suggests that the helical domain mutations and kinase domain mutations exert their oncogenic functions through distinct molecular mechanisms. p110α helical domain mutations alleviate the inhibitory interaction between p110α and the N-terminal SH2 domain (nSH2) of p85α and β, and facilitate direct interaction of p110α with IRS1[8–12]. The E545K helical domain mutation in p110α does not increase lipid kinase activity in response to phosphorylation of receptor tyrosine kinase or adaptor proteins[8,10–12]. Moreover, the p110α E545K helical domain mutant requires the Ras-binding domain to transform chicken embryonic fibroblasts[13], whereas the H1047R kinase domain mutant requires the p85-binding domain. Phenotypically, breast cancer cells expressing a *PIK3CA* helical domain mutation showed more severe metastatic phenotypes than that of wild-type or *PIK3CA* H1047R mutation in an isogenic background[14]. Furthermore, patients with a *PIK3CA* H1047R mutation have a better response to PI3K/AKT/mTOR inhibitors than patients with a *PIK3CA* helical domain mutation in early phase clinical trials[15,16].

The regulatory subunits of PI3K p85α (encoded by *PIK3R1*) and p85β (encoded by *PIK3R2*) are ubiquitously expressed, but they seem to play opposite roles in tumorigenesis[17]. Loss-of-function mutations of p85α frequently occur in endometrial and brain cancers[18,19], suggesting that it normally functions as a tumor suppressor[20,21]. In contrast, p85β often is overexpressed in diverse cancers and depletion of p85β impairs tumor formation in vivo and in vitro[22–26], suggesting that it plays an oncogenic role in tumorigenesis. Intriguingly, several studies have demonstrated that p85 proteins can regulate biological processes through nuclear translocation[27–31]. It has been shown that p85α and p85β interact with nuclear Bromodomain-containing protein 7 (BRD7) and X-box binding protein 1 (XBP1) to regulate glucose homeostasis[30,31]. However, it remains unknown whether nuclear p85 proteins play an important role in tumorigenesis.

Here we show that p85β, but not p85α, disassociates from p110α and translocates into the nucleus in cancer cells with a p110α helical domain mutation, thereby promoting tumor growth. The nuclear p85β recruits deubiquitinase USP7 to stabilize histone methyltransferases EZH1 and EZH2 and enhances histone H3 lysine 27 trimethylation (H3K27Me3). Moreover, we demonstrate that a combination of Alpelisib and an EZH inhibitor, Tazemetostat, induces regression of xenograft tumors

harboring a *PIK3CA* helical domain mutation, but not tumors with either a WT *PIK3CA* or a *PIK3CA* kinase domain mutation.

## Results

**p85β disassociates from p110α helical domain mutant protein.** We previously demonstrated that p110α helical domain mutant proteins (e.g. E545K) gain direct interaction with IRS1 independent of p85α and β, thereby rewiring oncogenic signaling[12]. To gain insights into how the p110α helical domain mutations impact PI3K complex formation, we immunoprecipitated either wild-type (WT) p110α or p110α E545K mutant protein in cell lines with the WT or mutant endogenous p110α proteins tagged with 3 × FLAG [12, Fig. 1a]. Interestingly, compared to the WT p110α protein, the p110α E545K protein drastically reduced its binding to p85β, but not p85α (Fig. 1b and S1a). This observation was further validated by reciprocal immunoprecipitation of either p85β or p85α in isogenic DLD1 cell lines with either WT-only (*PIK3CA* E545K allele knockout) or E545K-only (*PIK3CA* WT allele knockout, Fig. 1c, d and S1b, c). We postulated that some of the p85β proteins might disassociate from the PI3K complexes in *PIK3CA* E545K mutant cells. Indeed, immunoprecipitation with both p110α and p110β in the isogenic DLD1 E545K-only and WT-only cells showed more PI3K complex-free (post-IP) p85β were present in the E545K-only cells than in the WT-only cells (Figure S1d). These results were further validated by gel-filtration analyses (Figure S1e).

Tumor-derived *PIK3CA* mutations are clustered in two hotspots: one in the helical domain at E542, E545, Q546, and the other in the kinase domain at the H1047 site. We set out to determine which mutations in p110α affect binding to p85β. As shown in Fig. 1e, f and S1f, as with the p110α E545K mutant, the p110α E542K and Q546K mutant proteins disrupted their interactions with p85β compared to the WT p110α. In contrast, the kinase domain p110α H1047R mutant protein had no impact on p85β binding. Neither did other relatively rare p110α mutant proteins including R88Q and K111N in the ABD domain, N345K and C420R in the C2 domain, M1043I and G1049R in the kinase domain (Figs. 1e, f, and S1f).

We then used nine different cell lines to assess p85β-p110 complex formation. Consistently, the interaction between p85β and p110α were weaker in four cell lines with *PIK3CA* helical domain mutation (DLD1, H460, MB361, and SW948) than in either two cell lines with wild-type *PIK3CA* (SW480 and LoVo) or three cell lines with *PIK3CA* H1047R mutation (HCT116, T47D, and RKO) (Fig. 1g, h and Fig. S1g, i). In contrast, the interactions between p85β and p110β or between p85α and p110α were similar in these nine cell lines (Fig. 1g, h and Fig. S1g, i). Together, our data suggest that the p85β protein dissociates from the mutant p110α in cancer cells with a *PIK3CA* helical domain mutation.

**The N-terminal p85β sequences cause its dissociation from p110α helical domain mutant protein.** It is interesting that p85β, but not p85α, disassociates from the p110α helical domain mutant proteins. A protein sequence alignment showed that the SH3 domain, GAP domain, and link region between the GAP and nSH2 domains in the N-terminus sequences between p85α and p85β are less conserved (56% identical, Fig. 1i). In contrast, the three SH2 domains in the two proteins' C-terminus are highly conserved (91% identical, Fig. 1i). We postulated that the N-terminus sequences of p85β might cause its disassociation from the p110α helical domain mutant proteins. To this end, we constructed two chimeric p85 proteins that swapped the N-terminus regions of p85α and p85β (Fig. 1i). As shown in Fig. 1j, the N-terminal p85α-C-terminal p85β chimeric proteins

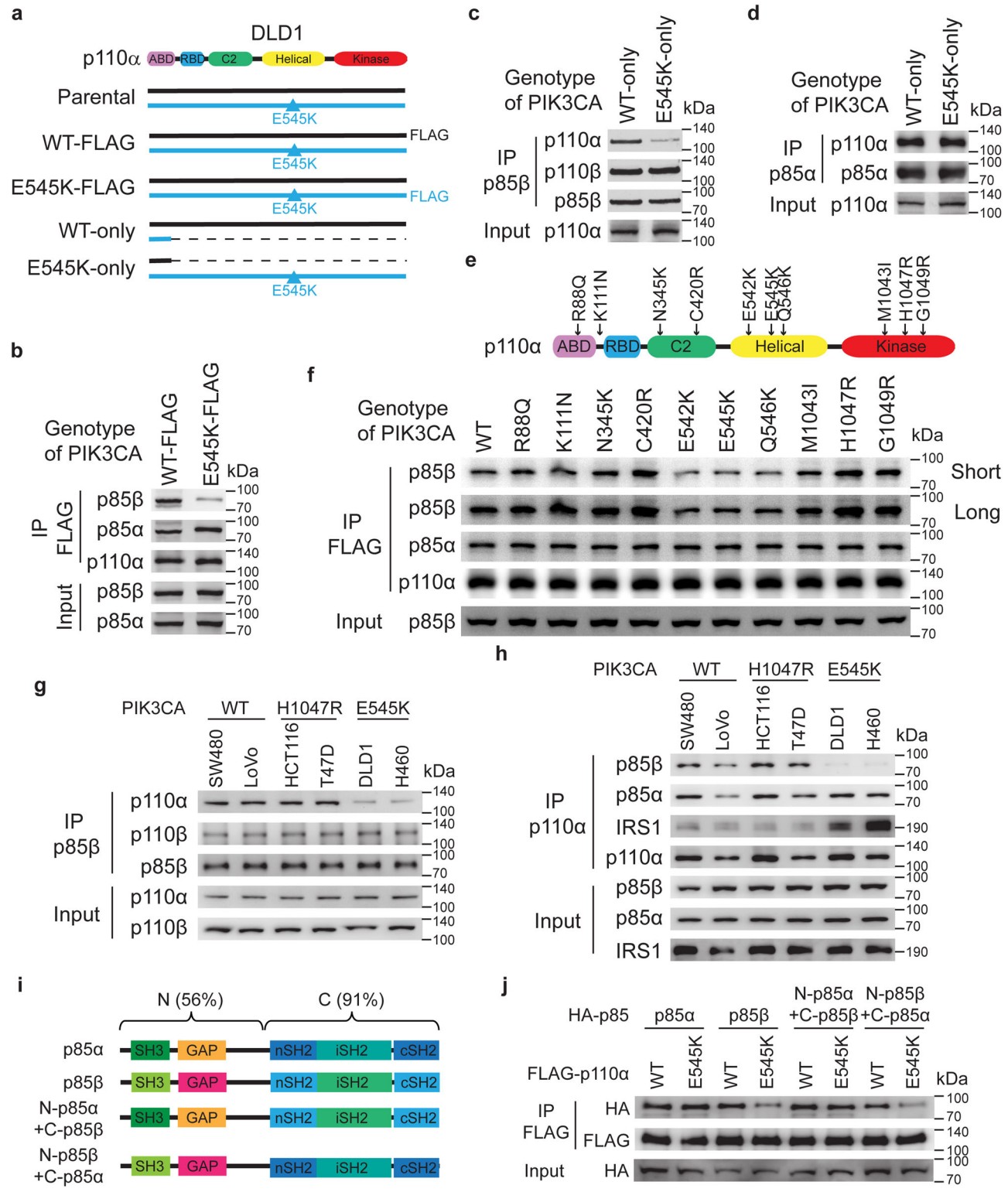

bound similarly to both p110α WT and E545K mutant proteins, whereas the N-terminal p85β–C-terminal p85α chimeric protein bound to less p110α E545K mutant protein compared to the WT counterpart (Fig. 1j). These data suggest that the N-terminal sequences of p85β cause its disassociation from the p110α helical domain mutant proteins.

**p85β plays an oncogenic role in cancer cells with a _PIK3CA_ helical domain mutation**. To explore the specific function of

p85β in p110α helical domain mutated tumors, we assessed whether p85β expression levels are associated with any clinical outcomes. We chose to analyze the following four TCGA datasets because _PIK3CA_ is frequently mutated, and PIK3R2 is over-expressed in these tumor types: colorectal cancer (COAD), bladder carcinoma (BLCA), endometrial carcinoma (UCEC), and breast cancer (BRCA) (Fig. S2a). Because the number of patients whose tumors harbor _PIK3CA_ mutations is small in each tumor type, we combined the TCGA data of the four tumor types

**Fig. 1 p85β disassociates from the p110α helical domain-mutant protein. a** Schematic of DLD1 isogenic cell lines. WT-FLAG: DLD1 cells with the endogenous wild-type p110α tagged with 3×FLAG; E545K-FLAG: DLD1 cells with the endogenous p110α E545K mutant protein tagged with 3×FLAG; WT-only: DLD1 cells with the *PIK3CA* E545K allele knocked out; E545K-only: DLD1 cells with the *PIK3CA* WT allele knocked out. ABD: adaptor-binding domain; RBD: Ras-binding domain; C2: C2 domain; helical: helical domain; kinase: kinase domain. **b–d** p85β, but not p85α, disassociates from p110α E545K mutant protein. Cell lysates from the p110α E545K or WT FLAG-tagged cells were immunoprecipitated with anti-FLAG antibody-conjugated beads and blotted with indicated antibodies (**b**). Cell lysates from the indicated cell lines were immunoprecipitated with either an anti-p85β antibody (**c**) or an anti-p85α antibody (**d**) and blotted with indicated antibodies. **e** A schematic diagram of tumor-derived *PIK3CA* mutations tested for interaction with p85β. **f** p85β, but not p85α, disassociates from p110α helical domain mutant proteins. The indicated FLAG-tagged p110α constructs were transfected into 293 T cells. Cell lysates were immunoprecipitated by anti-FLAG agarose and then blotted with the indicated antibodies. **g, h** p85β disassociates from PI3K complexes in *PIK3CA* helical domain mutant cells. Cell lysates from indicated cell lines were immunoprecipitated with either an anti-p85β antibody (**g**) or an anti-p110α antibody (**h**) and blotted with indicated antibodies. **i, j** The N-terminal domains of p85β cause disassociation from p110α E545K mutant protein. Schematics of p85α, p85β, and two chimeric p85 constructs (**i**). The indicated HA-tagged p85 constructs were co-transfected with a Flag-tagged construct expressing either WT or E545K mutant p110α. Cell lysates were immunoprecipitated by anti-FLAG agarose and then blotted with an anti-HA antibody. Similar results were reproduced three times.

together and divided the patients into three groups according to the *PIK3CA* mutation status: helical domain mutation group, non-helical domain mutation group, and wild-type group. Interestingly, high expression of *PIK3R2* was found to be significantly associated with poor five-year survival only in the helical domain mutation group (HR = 2.146, 95% CI 1.162~3.96, $p = 0.0148$), but not in the non-helical domain mutation group (HR = 0.9126, 95% CI 0.5714~1.4, $p = 0.7017$) or wild-type group (HR = 0.9325, 95% CI 0.7404~1.1, $p = 0.5527$) (Fig. 2a). These data suggest that p85β may promote tumorigenesis in tumors with a *PIK3CA* helical domain mutation.

**The depletion of p85β reduces the growth of cancer cells with a *PIK3CA* helical domain mutation, but not cells with WT or a kinase domain mutant *PIK3CA*.** To further investigate the function of p85β in the context of *PIK3CA* helical domain mutation, we knocked out p85β in the isogenic DLD1 *PIK3CA* E545K-only and *PIK3CA* WT-only cell lines (Fig. 2b). As shown in Fig. 2c–e, knockout of p85β reduced cell proliferation, colony formation, and xenograft tumor growth of the DLD1 *PIK3CA* E545K-only cells, but not the WT *PIK3CA*-only counterpart. Similarly, knockout of p85β in H460 cells, which harbor a *PIK3CA* E545K mutation, reduced cell proliferation, colony formation, and xenograft tumor growth (Fig. 2f–i). But knockout of p85β in RKO cells, which harbor a *PIK3CA* kinase domain mutation, had no impact on tumor growth (Fig. 2f–i). Furthermore, knockdown of p85β reduced cell proliferation and colony formation of *PIK3CA* E545K mutant MDA-MB361 breast cancer cells and *PIK3CA* E542K mutant SW948 cells (Fig. S2b, d). In contrast, knockdown of p85β had no impact on cell proliferation and colony formation of *PIK3CA* H1047R mutant T47D breast cancer cells and *PIK3CA* WT SW480 colon cancer cells (Figure S2e–h). Together, these data suggest that p85β promotes the growth of tumors harboring *PIK3CA* helical domain mutations, but not those tumors with WT or *PIK3CA* kinase domain mutations.

**p85β translocates into the nucleus in cancer cells with a *PIK3CA* helical domain mutation.** Next, we set out to elucidate the molecular mechanisms by which p85β promotes the growth of cancer cells with a *PIK3CA* helical domain mutation. Given that it is well documented that PI3K transduces signaling to AKT, we examined whether p85β knockout impacts AKT and its downstream signaling. As shown in Fig. S3a, knockout of p85β did not affect phosphorylation of AKT, GSK3β, Foxo, mTOR, and p70S6K in DLD1 *PIK3CA* E545K-only cells regardless of whether under serum starvation conditions or when stimulated by insulin or EGF. Moreover, neither knockout of p85β impacted p110α and

p110β protein levels in *PIK3CA* E545K-only cells (Fig. S3a), nor did overexpression of p85β affect the levels of p110α, p110β, and AKT phosphorylation in DLD1 *PIK3CA* E545K-only cells (Fig. S3b). In contrast, overexpression of p85α increases the levels of p110α, p110β, and AKT phosphorylation (Fig.e S3b). Those data suggest that p110 proteins are stabilized by p85α in DLD1 cells.

It has been reported that p85 proteins can translocate to the nucleus[27–31]. We thus performed immunofluorescent staining of p85β in the isogenic DLD1 *PIK3CA* E545K-only and *PIK3CA* WT-only cell lines. As shown in Fig. 3a and Fig. S3c, p85β was present in both the nucleus and cytoplasm in the DLD1 *PIK3CA* E545K-only cells, but only in the cytoplasm in the DLD1 *PIK3CA* WT-only cells. The specificity of the anti-p85β antibody was demonstrated by the lack of staining in the DLD1 *PIK3CA* E545K-only p85β knockout cells (Fig. 3a). Consistently, cell fractionation analyses showed that significantly more p85β was located in the nucleus in the DLD1 *PIK3CA* E545K-only cells than in the DLD1 *PIK3CA* WT-only cells (Fig. 3b, c). In agreement with previous reports, a small fraction of p85α was also present in the nucleus (Fig. 3b), although the amounts of nuclear p85α were similar between the isogenic DLD1 *PIK3CA* E545K-only cells and DLD1 *PIK3CA* WT-only cells (Fig. 3c). Because BRD7 was reported to facilitate nuclear translocation of p85α[27–31], we compared BRD7 protein levels between the DLD1 *PIK3CA* E545K-only and *PIK3CA* WT-only cells. Figure 3b shows that BRD7 proteins were largely localized to the nucleus to a similar degree in the two cell lines, suggesting that the differential nuclear localization of p85β in the DLD1 *PIK3CA* E545K cells is unlikely to be mediated by BRD7. Consistently, knockout of BRD7 in DLD1 cells largely abolished nuclear translocation of p85α but not p85β (Fig. S3d, e).

To assess the generality and specificity of p85β nuclear localization in *PIK3CA* helical domain mutant cells, we analyzed cellular localization of p85β in a panel of cell lines: two additional *PIK3CA* E545K mutant cell lines (H460 and MB361) and a *PIK3CA* E542K helical domain mutant cell line SW948; three *PIK3CA* H1047R mutant cells lines (HCT116, RKO, and T47D); and two WT *PIK3CA* cell lines (SW480 and LoVo). Both immunofluorescent staining and cell fractionation demonstrated that p85β was translocated into the nucleus in the cell lines with a *PIK3CA* E545K mutation, but not in cell lines with WT *PIK3CA* or H1047R mutation (Fig. 3d–j). Furthermore, overexpressing *PIK3CA* E545K protein into SW480 cells facilitates the nuclear translocation of p85β (Fig. S3e). Consistently, immunohistochemistry staining of human colon cancer specimens showed that p85β was predominantly localized in the nuclei of tumors with a *PIK3CA* E545K mutation (Fig. S3g), but mostly localized in the cytoplasm of tumors with wild-type *PIK3CA* or a *PIK3CA*

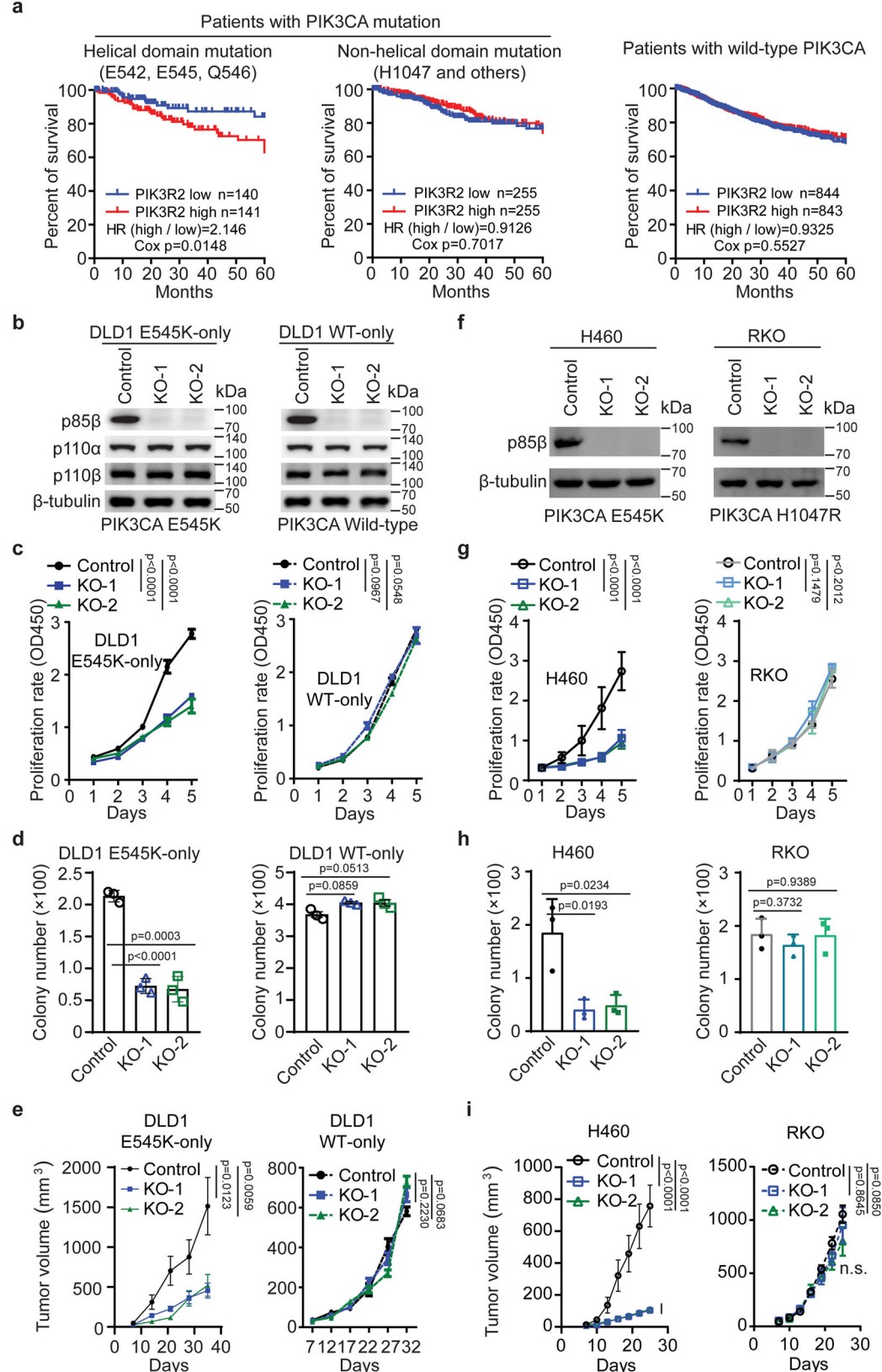

H1047R mutation (Fig. S3g). Last, live-cell imaging also showed that p85β-mCherry initially co-localized with p110α E545K-GFP (yellow color), then dissociated from the mutant p110 E545K-GFP and translocated into the nucleus (supplementary video 1, Fig. S3h), whereas p85β-mCherry did not dissociate from WT p110α-GFP (supplementary video 2, Fig. S3h) or p110α H1047R-GFP fusion protein (supplementary video 3, Fig. S3h). Together, those data suggest that p85β dissociates from the PI3K complexes and translocates into the nucleus in cancer cells with a *PIK3CA* helical domain mutation.

**Fig. 2 p85β plays an oncogenic role in cancer cells with *PIK3CA* helical domain mutations. a** High levels of *PIK3R2* (p85ß) are associated with worse survival of patients whose tumors harbor a *PIK3CA* helical domain mutation. COAD, BLCA, UCEC, and BRAC datasets were downloaded from TGCA and combined. Patients were divided into three groups according to their *PIK3CA* mutation status: Helical domain mutations, Non-helical domain mutations, and wild-type. Kaplan-Meier analyses of 5-year survival of patients whose tumors expressing high levels of *PIK3R2* vs low levels of *PIK3R2* were performed. HR: Hazard Ratio. **b**–**i** Knockout of p85β impairs the growth of cancer cells with a *PIK3CA* E545K mutation, but not cells with WT *PIK3CA* or a *PIK3CA* H1047R mutation. *PIK3R2* was knocked out in the indicated cell lines, and the cells were assayed for: Western blot analyses of p85β, p110α, and p110β proteins (**b**, **f**); cell proliferation (**c**, **g**); colony formation (**d**, **h**); and xenograft tumor (10 tumors/group) growth (**e**, **i**). Statistical analyses, two-way ANOVA was used for **c**, **e**, **f** and **h**, and student's *t*-test (two-tailed) was used for **d**, **i**. Data are presented as mean ± SEM of three independent experiments. Source data are provided as a Source Data file.

**Nuclear translocation of p85β is critical for tumorigenicity of *PIK3CA* E545K mutant cells.** To determine how p85β translocates into the nucleus, we exploited a nuclear localization sequence (NLS) prediction tool (cNLS Mapper) and identified a putative NLS at amino acids 474 to 484 of p85β (Fig. 4a). To test if this NLS mediates nuclear translocation of p85β, we first fused the NLS sequence with GFP and showed that the p85βNLS-GFP was localized only in the nucleus (Figure S4a). In contrast, the p85βNLS$^{KR-AA}$-GFP, in which the two basic amino acids K477 and R478 in the NLS were mutated to alanine, was diffused in cells just as the GFP alone (Fig. S4a). We then reconstituted the DLD1 *PIK3CA* E545K-only p85β KO cells with HA-tagged WT p85β, or mutant p85β construct (Fig. S4b). Although both the WT and K477A/R478A mutant p85β bound similarly to WT p110α and p110β, immunofluorescent staining and cell fractionation showed that the WT p85β translocated into the nucleus, but the K477A/R478A mutant p85β remained in the cytoplasm (Figures S4c, d). Interestingly, reconstitution of the WT, but not the mutant p85β, rescued the defects in cell growth and colony formation of the DLD1 *PIK3CA* E545K-only p85β KO cells (Figs. S4e, f). As expected, both WT and the mutant p85β reduced their interactions with p110α E545K mutant protein, but retained the interactions with p110α H1047R mutant protein and p110β (Fig. S4b, g). To further validate this observation, we generated p85β K477A/R478A mutant knockin (KI) in DLD1 parental cells, which harbor a *PIK3CA* E545K mutation, using CRISPR/Cas9 mediated gene editing (Fig. 4b). Two independently-derived homozygous KI clones termed p85β$^{KR-AA}$ were chosen for in-depth analyses. The p85β$^{KR-AA}$ mutation did not impact levels of itself, p110α, p110β, and AKT phosphorylation (Fig. 4c), suggesting that the p85β NLS mutant does not affect the kinase activity of PI3K. Nonetheless, the NLS mutant p85β$^{KR-AA}$ failed to translocate into the nucleus (Fig. 4d). Moreover, p85β$^{KR-AA}$ mutant KI cell lines displayed reduced cell proliferation, colony formation, and xenograft tumor growth (Fig. 4e–g). Together, those data suggest that nuclear but not cytoplasmic p85β promotes the growth of cancer cells with a *PIK3CA* helical domain mutation.

**Nuclear p85β stabilizes EZH1/2 proteins, thereby increasing H3K27 tri-methylation.** We postulated that the nuclear p85β might regulate gene expression. Thus, we performed RNA-seq analyses of DLD1 parental cells and p85β$^{KR-AA}$ mutant cells. Compared with the parental cells, expression levels of 3,224 genes were up-regulated, and 2,044 genes were down-regulated in the p85β$^{KR-AA}$ mutant cells (Fig. 5a). Interestingly, many of the well-known tumor suppressor genes, such as APC, ATM, BRCA1/2, SMAD4, were up-regulated in p85β$^{KR-AA}$ mutant cells (Fig. S5a, b). These data suggest that the nuclear p85β might regulate global gene transcription. We thus examined histone modifications in parental cells and p85β$^{KR-AA}$ mutant clones. As shown in Fig. 5b, levels of histone H3K27 trimethylation (H3K27me3) were reduced in the two p85β$^{KR-AA}$ mutant clones compared to the parental DLD1 cells (Fig. 5b). Consistently, levels of H3K27me3 were higher in DLD1 *PIK3CA* E545K-only cells than in the

isogenic *PIK3CA* WT-only cells (Fig. 5c). Moreover, knockout of p85β in DLD1 *PIK3CA* E545K-only cells decreased the levels of H3K27me3 (Fig. 5c), whereas knockout of p85β in DLD1 *PIK3CA* WT-only cells had no impact on H3K27me3 (Fig. 5c). In contrast, knockout of p85β did not affect histone trimethylation at other sites, including H3K4, H3K9, H3K36, and H3K76 in either DLD1 *PIK3CA* E545K-only or DLD1 *PIK3CA* WT-only cells (Fig. 5c). Furthermore, the reconstitution of WT p85β, but not p85β$^{KR-AA}$ mutant, in DLD1 *PIK3CA* E545K-only p85β KO cells restored the levels of H3K27me3 (Fig. S5c). To assess the generality of those observations, we knocked down p85β in three additional *PIK3CA* helical domain mutant cell lines (H460, MB-361, and SW948) and showed that knockdown of p85β in those cell lines reduced the levels of H3K27me3 (Fig. 5d). In contrast, knockdown of p85β in cell lines with a WT *PIK3CA* (SW480) or a *PIK3CA* H1047R mutation (HCT116, RKO, and T47D) did not impact the levels of H3K27me3 (Fig. 5e). Taken together, the data suggest that nuclear p85β modulates H3K27me3, a marker for transcriptional repression.

Given that EZH1 and EZH2 are the histone methyltransferases for the H3K27 site, we next examined if nuclear p85β regulates EZH1 and EZH2. As shown in Fig. 5b, compared to parental cells, levels of EZH1 and EZH2 proteins were markedly reduced in p85β$^{KR-AA}$ mutant knockin clones. Consistently, levels of EZH1 and EZH2 proteins were higher in DLD1 *PIK3CA* E545K-only cells than in the isogenic *PIK3CA* WT-only cells (Fig. 5c). The depletion of p85β decreased EZH1 and EZH2 protein levels in *PIK3CA* helical domain mutant cell lines, including H460, MB-361, and SW948 (Fig. 5d). In contrast, Fig. 5e shows that the depletion of p85β did not impact EZH1 and EZH2 protein levels in cell lines with WT *PIK3CA* (SW480) or a *PIK3CA* H1047R mutation (HCT116, RKO, and T47D). Conversely, the reconstitution of WT p85β, but not p85β$^{KR-AA}$ mutant, in DLD1 *PIK3CA* E545K-only p85β KO cells restored EZH1 and EZH2 protein levels (Fig. S5c). Notably, the knockout of p85β in DLD1 *PIK3CA* E545K-only cells did not impact the mRNA expression of EZH1 and EZH2 (Fig. S5d). Consistently, the half-life times of EZH1 and EZH2 proteins were reduced in p85β KO cells compared to the parental counterparts (Fig. S5e, f). It is worth noting that the knockout of p85β did not impact other components of the PRC2 complex (Fig. 5c). Together, these data suggest that nuclear p85β regulates EZH1 and EZH2 protein stability, thereby enhancing H3K27me3.

**The nuclear p85β impacts H3K27me3 at specific genome regions.** To assess how p85β impacts chromatin-associated H3K27me3, we performed ChIP-seq analyses of H3K27me3 in DLD1 parental and p85β$^{KR-AA}$ mutant cells. As shown in Fig. 5, compared to the parental DLD1 cells, the binding of H3K27me3 was reduced in approximately half of the chromatin regions in p85β$^{KR-AA}$ mutant cells (Fig. 5f). Specific analysis of the promoter regions showed that promoter-associated H3K27me3 in 383 genes were decreased in p85β$^{KR-AA}$ mutant cells compared to the parental counterpart. Integrated analyses with the RNA-seq data

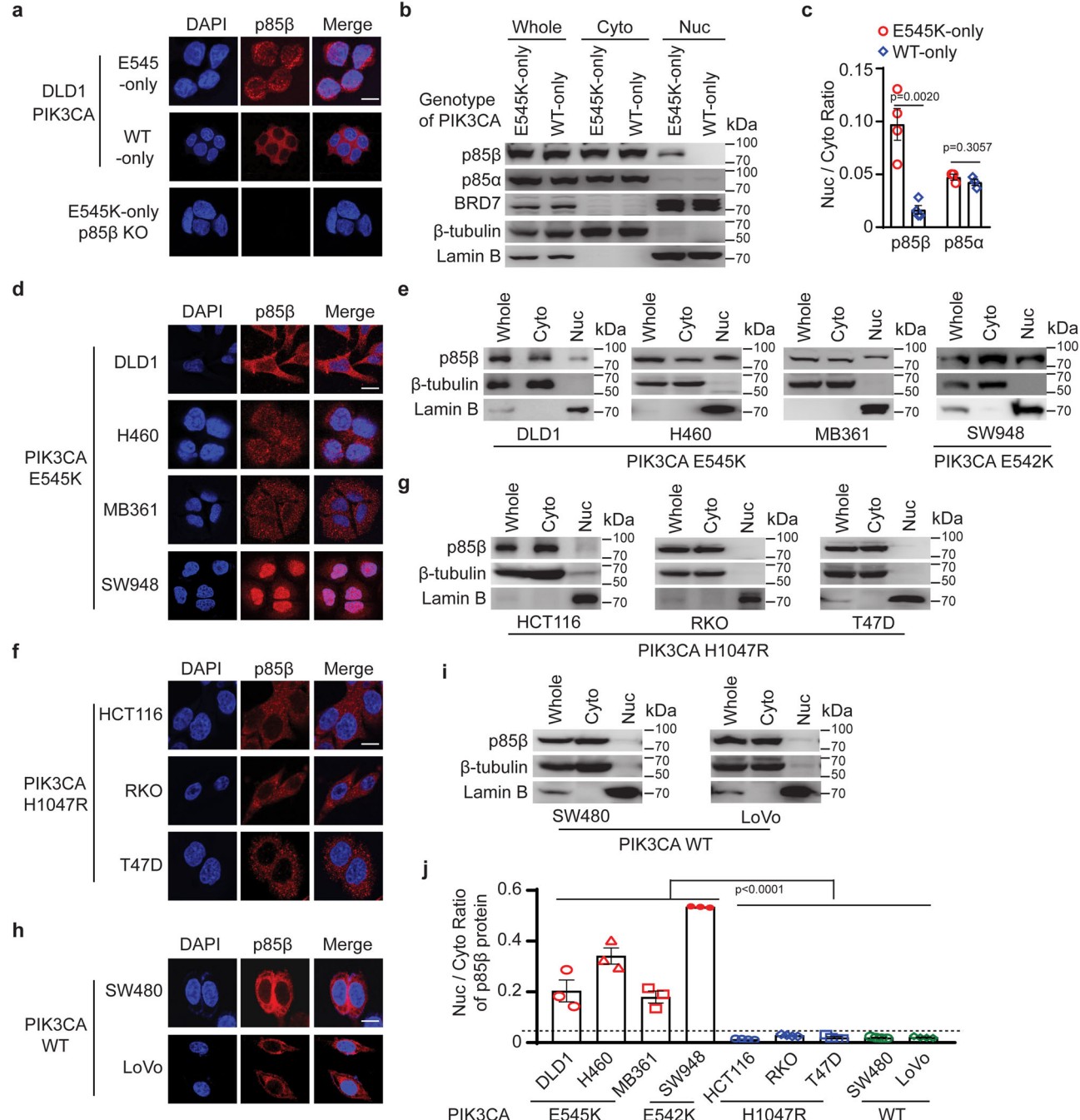

**Fig. 3 p85β translocates into the nucleus in cancer cells with a *PIK3CA* E545K mutation. a–c** p85β translocates into the nucleus in DLD1 *PIK3CA* E545K cells. **a** The indicated cells were immunofluorescently stained with an anti-p85β antibody and DAPI. **b** Cell lysates were fractionated into cytoplasmic (Cyto) and nuclear (Nuc) fractions and blotted with the indicated antibodies. Whole: whole cell lysate. The ratios of nuclear/cytoplasmic p85β levels were quantified by Image J as shown in (**c**). Data are presented as mean ± SEM of three (p85α) or four (p85 β) independent experiments. **d–j** p85β translocates into the nucleus in cancer cells with a *PIK3CA* helical domain mutation, but not cells with WT *PIK3CA* or a *PIK3CA* kinase domain mutation. The indicated cells were immunofluorescently stained with an anti-p85β antibody and representative images are shown in (**d**), (**f**), and (**h**). Cell lysates of the indicated cells were fractionated into cytoplasmic and nuclear fractions and blotted with the indicated antibodies (**e**), (**g**), and (**i**). The ratios of nuclear/cytoplasmic p85β levels were quantified by Image J and shown in (**j**). Data are presented as mean ± SEM of three independent experiments. The student's *t*-test (two-tailed) was used for statistical analyses. Source data are provided as a Source Data file. Scale bars = 10 μm.

indicated that the promoter regions of 38 genes (Table S1), whose RNA expression levels were up-regulated in p85β[KR-AA] mutant cells compared to parental DLD1 cells, had less enrichment for H3K27me3 in p85β[KR-AA] mutant cells than in parental DLD1 cells. We chose to validate DLG2 because it is the top-ranked gene among the 38 genes. Moreover, DLG2 has been

shown to have tumor suppressor function[32,33]. Our ChIP-PCR analyses showed that the enrichment of H3K27me3 in the promoter region of DLG2 was indeed significantly reduced in p85β[KR-AA] mutant cells compared to parental DLD1 cells (Fig. S5g).

Given that the EZH containing PRC2 complexes are predominantly located in the heterochromatin regions. We

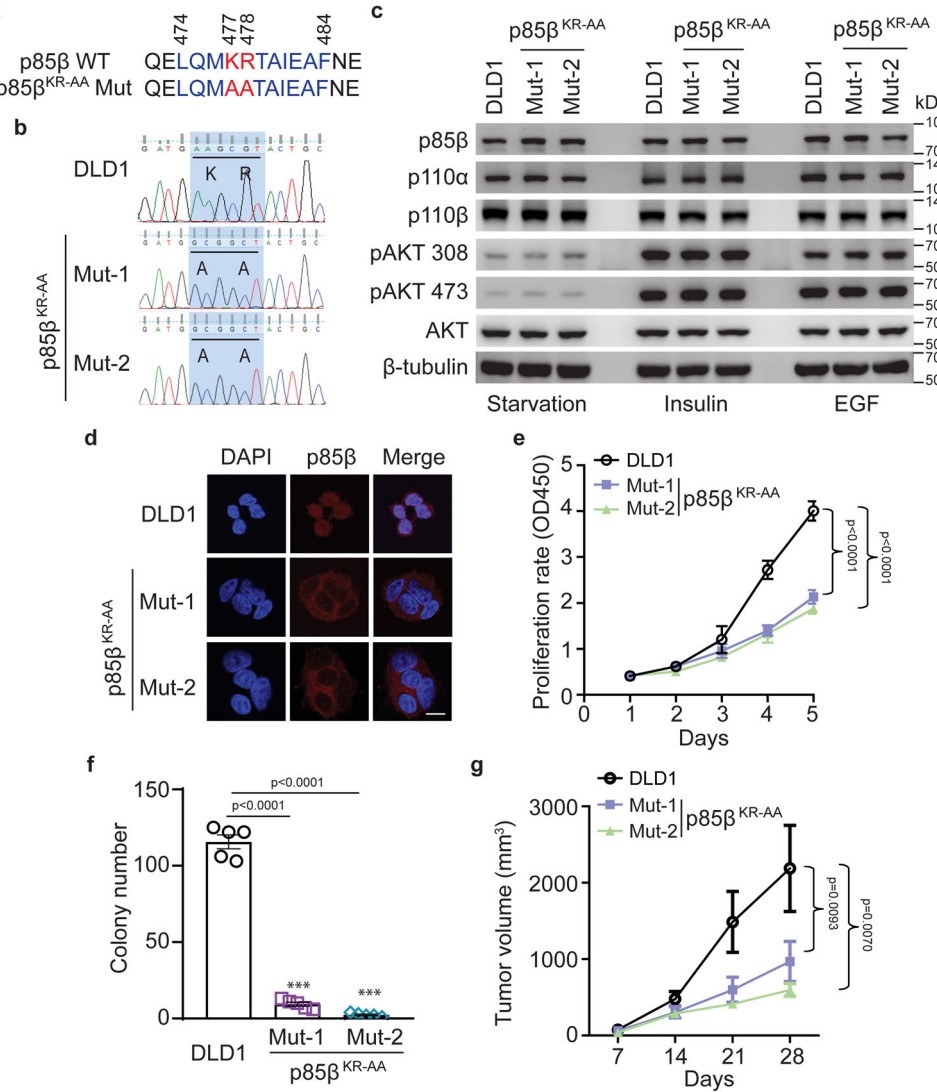

**Fig. 4 Nuclear translocation of p85β is critical for the tumorigenicity of *PIK3CA* E545K mutant cells. a** A predicted Nuclear Localization Sequence (NLS) in p85β protein is highlighted in blue. The critical stretch basic amino acids $K^{477}R^{478}$ are highlighted in red. **b** Genomic DNA sequencing of DLD1 parental cells and K477A R478A mutant knock-in (p85β$^{KR-AA}$) cells. **c** NLS mutation has no impact on p85β, p110α and p110β protein levels and AKT phosphorylation. Cells of the indicated genotypes were serum-starved, stimulated with insulin or EGF, and then lysed and blotted with indicated antibodies. **d** Cells of the indicated genotype were stained with an anti-p85β antibody. Mut-1 and Mut-2 are two independently derived p85β$^{KR-AA}$ mutant knock-in clones. **e–g** Cells of the indicated genotypes were assayed for cell proliferation (**e**), colony formation ($n = 5$) (**f**), and xenograft tumor growth (10 tumors/group) (**g**). Statistical analyses, two-way ANOVA was used for **e**, **g**, and student's *t*-test (two-tailed) was used for **f**. Data are presented as mean ± SEM of three independent experiments. Source data are provided as a Source Data file. Scale bar = 10 μm.

performed H3K27me3 ChIP-PCR on seven repeat sequences in different genomic regions in parental DLD1 and p85β$^{KR-AA}$ mutant cells. Compared to the parental cells, the binding of H3K27me3 to D4Z4 repeat sequences on chromosome 4, satellite sequences on chromosome 1, Alu sequences on chromosome 19, and the telomeric TTAGGC repeats at the chromosome 7q was drastically reduced in p85β$^{KR-AA}$ mutant cells (Fig. S5h), whereas the binding of H3K27me3 to satellite sequences on chromosome 4 and Alu sequences on chromosome 10 was decreased modestly (Fig. S5h). In contrast, the binding of H3K27me3 to the promoter region of testis-specific histone 2B was not changed between parental DLD1 and p85β$^{KR-AA}$ mutant cells. We postulate that the amount of p85β binding to different chromatin regions varies, thereby modulating EZH and H3K27me3 differentially. Together, the data suggest that p85β impacts chromatin-bound H3K27me3 at specific genome regions.

**Nuclear p85β recruits USP7 to EZH1/2 to protect them from ubiquitin-mediated protein degradation.** We next set out to determine how nuclear p85β stabilizes EZH1 and EZH2 proteins. It has been reported that USP7 deubiquitinates and stabilizes EZH2 in prostate cancer cells[34,35]. We thus postulated that nuclear p85β brings USP7 to EZH1 and EZH2, thereby protecting them from ubiquitin-mediated protein degradation. Indeed, immunoprecipitation analyses showed that p85β interacted with USP7, EZH1, and EZH2 in cell lines harboring a *PIK3CA* E545K mutation, including DLD1, MB361, and H460 (Fig. 6a, b, Fig. S6a, b). In contrast, p85β did not interact with USP7, EZH1, or EZH2 in *PIK3CA* WT (SW480) and H104R mutant (HCT116, T47D, and RKO) cell lines (Fig. 6b and Fig. S6b). Knockout or knockdown of p85β reduced the binding of USP7 to EZH1 or EZH2 in *PIK3CA* E545K mutant cell lines DLD1 (Fig. 6c) and MB361 (Fig. 6d). Moreover, the knockdown of p85α had no

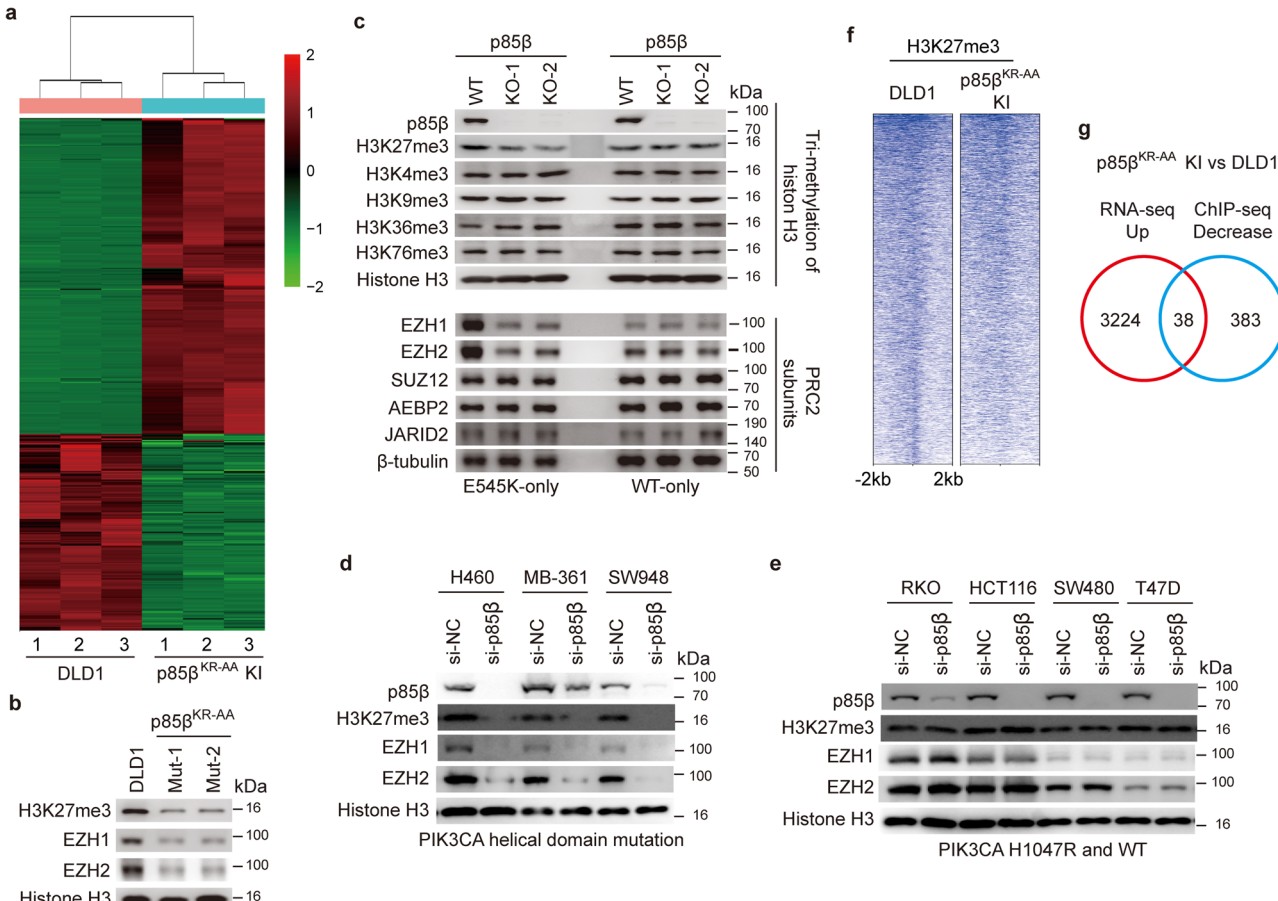

**Fig. 5 Nuclear p85β stabilizes EZH1/EZH2 and enhances H3K27me3 to regulate gene expression. a** A heat map of RNA-seq analyses of DLD1 parental cells and p85β NLS mutant (p85β^KR-AA KI) cells. **b–e** Nuclear p85β stabilizes EZH1 and EZH2 and increases histone H3K27 methylation in *PIK3CA* helical domain mutant cells, but not in *PIK3CA* WT or H1047R mutant cells. H3K27me3, EZH1, and EZH2 levels were evaluated by Western blot analyses in indicated cell lines. Similar results were reproduced three times. **f, g** ChIP-seq analyses of H3K27me3 in DLD1 parental cells and p85β NLS mutant cells. A heat map is shown in (**f**), and a Venn diagram is shown in (**g**).

impact on the interactions between EZH1/2 and USP7 (Fig. 6e). Together, the data suggest that p85β brings USP7 to the EZH1/2 complexes. Furthermore, compared to the parental cells, levels of EZH1 and EZH2 proteins and H3K27me3 (Fig. 6f) were reduced in DLD1 USP7 KO cells that we generated previously[36]. Consistently, proteasome inhibitor MG132 treatment restored EZH1 and EZH2 protein levels in USP7 KO cells (Fig. 6g), whereas EZH1 and EZH2 ubiquitination levels were increased in USP7 KO cells compared to the parental cells (Fig. 6h). Moreover, the stabilization of EZH1 and EZH2 by the nuclear p85β seems not to involve the p110α or p110β catalytic subunits, because EZH1 and EZH2 bound to p85β, but not p110α and p110β (Figure S6a). Taken together, the data suggest that nuclear p85β recruits USP7 to stabilize EZH1/2, thereby enhancing H3K27 trimethylation.

**A combination of an EZH inhibitor and the p110α inhibitor Alpelisib induces tumor repression.** We have shown that the nuclear p85β stabilizes EZH1 and EZH2 in cancer cells with a *PIK3CA* E545K mutation. Given that the mutation also activates p110α kinase activity, we hypothesized that a combination of an EZH inhibitor and a p110α inhibitor would have a better tumor inhibitory effect than either alone. We first tested a combination of p110α inhibitor Alpelisib (BYL-719) with an EZH inhibitor, GSK2816126. As shown in Fig. 7a, the drug combination induced tumor regression of xenografts established from DLD1 cells,

which harbor a *PIK3CA* E545K mutation, whereas single drugs alone only slowed tumor growth (Fig. 7a). Similar results were obtained with a combination of Alpelisib with another EZH2 inhibitor Tazemetostat (Fig. 7b). We chose Tazemetostat for in-depth studies, because it was recently approved by the FDA to treat EZH2 mutant follicular lymphoma and advanced epithelioid sarcoma[37]. Our hypothesis predicts that tumors that harbor a *PIK3CA* helical domain mutation are more sensitive to the drug combination than tumors with WT *PIK3CA*. To test this notion, we treated tumors established from either *PIK3CA* E545K-only or *PIK3CA* WT-only cells. The combination of Alpelisib and Tazemetostat induced regression of tumors established with *PIK3CA* E545K-only (Fig. 7c), whereas the drug combination only slowed down the growth of the *PIK3CA* WT-only tumors (Fig. 7d). Moreover, the drug combination did not induce tumor regression of a CRC patient-derived xenograft (PDX) harboring a *PIK3CA* H1047R kinase domain mutation (Fig. 7e). As shown in Fig. 2a, the *PIK3CA* helical domain mutations occurred in three residues (E545, E542, and Q546). Since we had demonstrated that the combination of Alpelisib and Tazemetostat induced tumor regression of *PIK3CA* E545K mutant tumors, we next tested if the drug combination induced regression of tumors harboring the other two recurrent *PIK3CA* helical domain mutations (E542K and Q546P). As shown in Fig. 7f, g, the combination of Alpelisib and Tazemetostat induced tumor regression of Vaco481 CRC cells with a *PIK3CA* Q546P mutation and a CRC PDX harboring

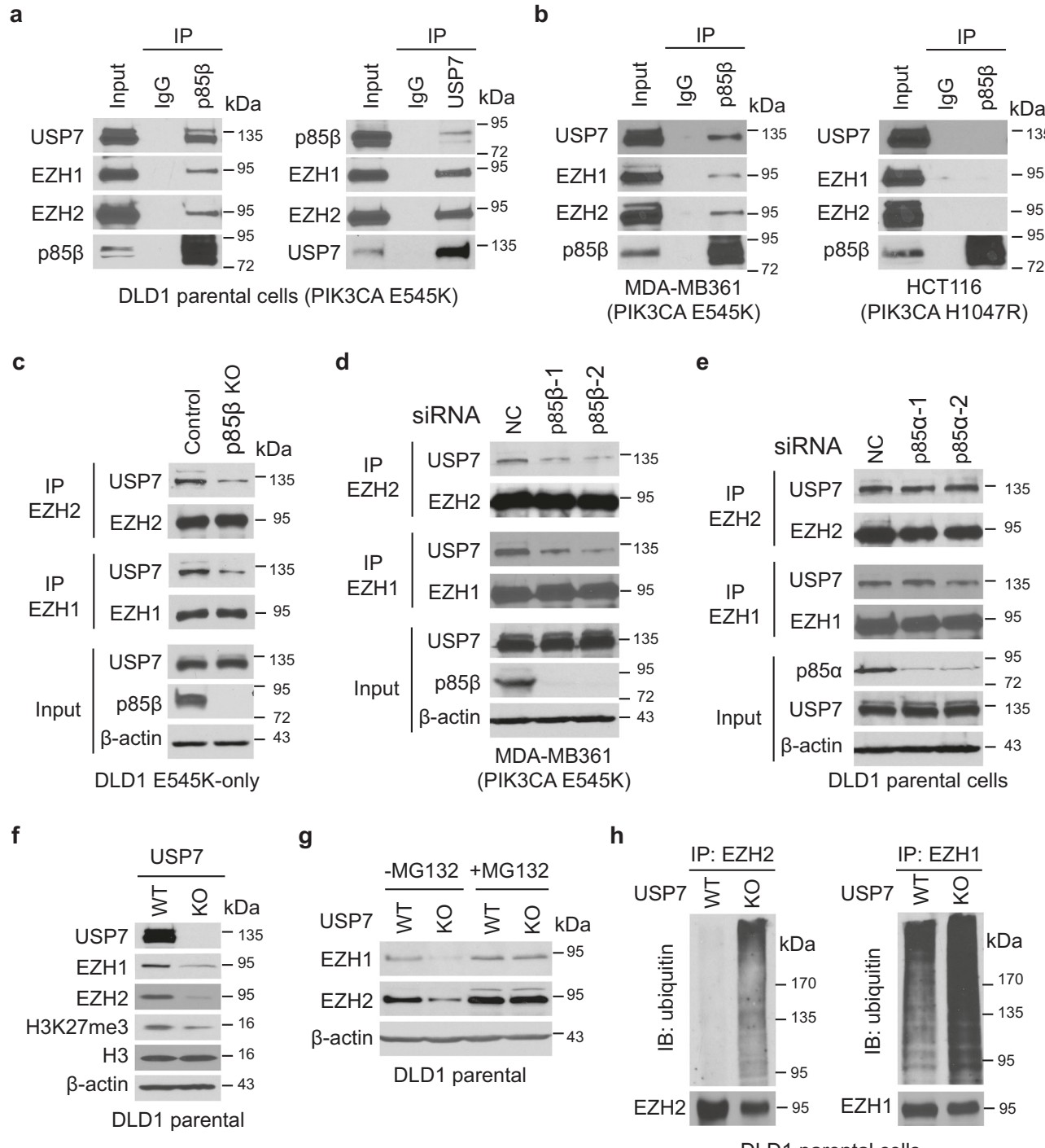

**Fig. 6 Nuclear p85β recruits the deubiquitinase USP7 to stabilize EZH1/2, increasing H3K27 tri-methylation. a** Nuclear p85β interacts with USP7, EZH1, and EZH2. DLD1 *PIK3CA* E545K cells were lysed and immunoprecipitated (IP) with either an anti-p85β or an anti-USP7 antibody, then blotted with indicated antibodies. **b** Nuclear p85β interacts with USP7, EZH1, and EZH2 in *PIK3CA* E545K mutant cells, but not in *PIK3CA* H1047R mutant cells. **c** The interaction between USP7 and EZH1 or EZH2 is reduced in p85β knockout cells. The indicated cell lines were lysed, IPed with either EZH1 or EZH2, then blotted with indicated antibodies. **d** The interaction between USP7 and EZH1 or EZH2 is reduced in p85β knockout *PIK3CA* E545K mutant MB361 cells. MB361 cells were transfected with scrambled siRNA (NC) or two independent siRNA against p85β. Cell lysates were IPed with either EZH1 or EZH2, then blotted with indicated antibodies. **e** Knockdown of p85α does not impact the interaction between USP7 and EZH1 or EZH2. DLD1 *PIK3CA* E545K cells were transfected with scrambled siRNA (NC) or two independent siRNA against p85α. Cell lysates were IPed with either EZH1 or EZH2, then blotted with indicated antibodies. **f–h** Deubiquitinase USP7 protects EZH1 and EZH2 from ubiquitin-mediated degradation. Lysates of DLD1 parental cells and USP7 knockout cells were blotted with indicated antibodies (**f**). DLD1 parental cells and USP7 knockout cells were treated with either vehicle or MG132 for 6 h and then blotted with indicated antibodies (**g**). After being treated with MG132 for 6 h, DLD1 parental cells and USP7 knockout cells were lysed, IPed with an antibody against either EZH1 or EZH2, then blotted with indicated antibodies (**h**). Similar results were reproduced three times.

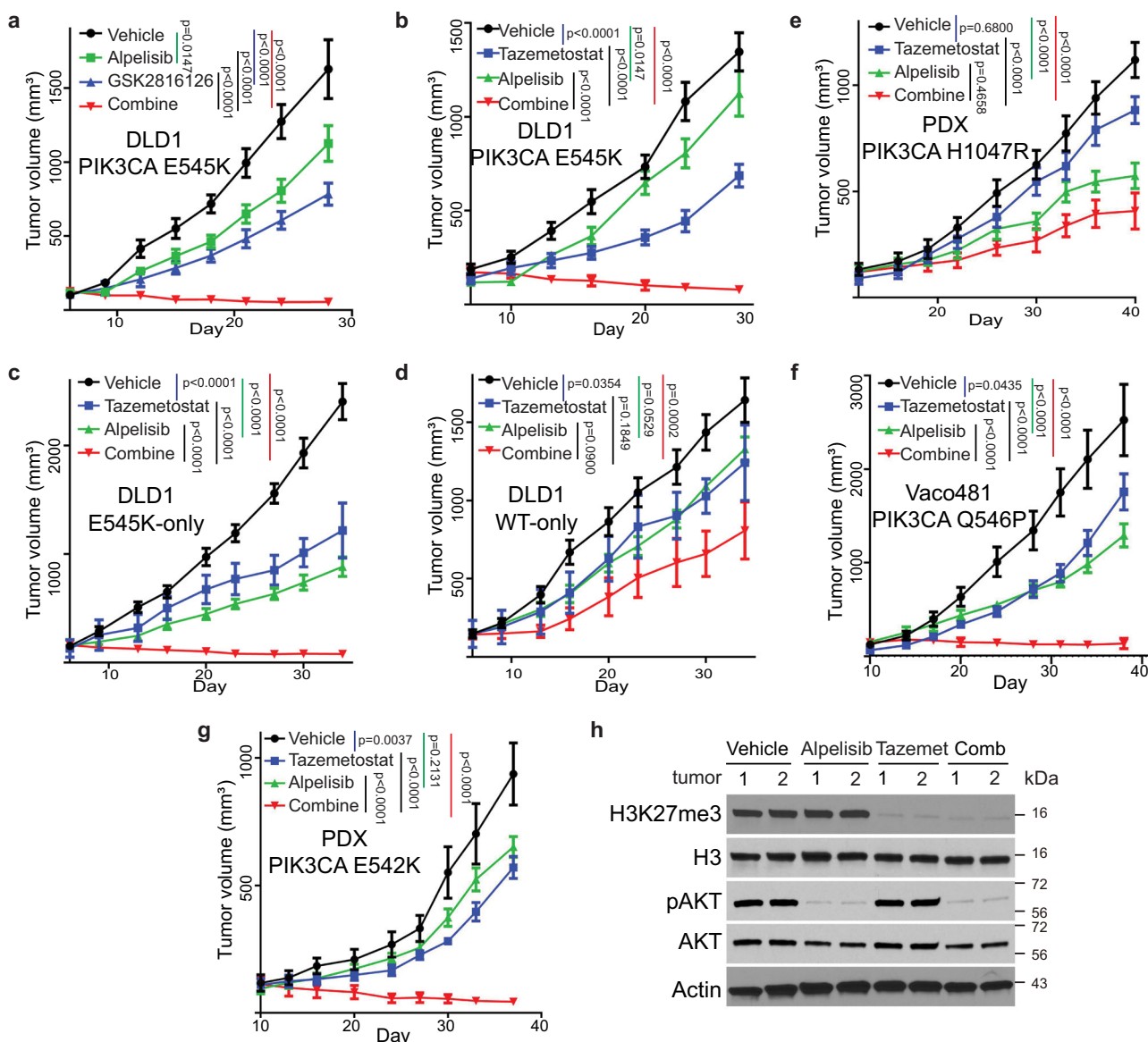

**Fig. 7 A combination of p110α and EZH inhibitors induces regression of tumors with *PIK3CA* helical domain mutations, but not WT or kinase domain mutation. a** Subcutaneous xenograft tumors established from DLD1 cells were treated with vehicle or the indicated drugs. GSK2814126: an EZH2 inhibitor; Alpelisib: a p110α-specific inhibitor. **b–g** Tumors are treated with an EZH inhibitor Tazematostat (EPZ-6438), Alpelisib, or the drug combination. Subcutaneous xenograft tumors established from DLD1 parental cells (**b**), DLD1 *PIK3CA* E545K-only cells (**c**), DLD1 *PIK3CA* WT-only cells (**d**); a CRC patient-derived xenograft (PDX) with a *PIK3CA* H1047R kinase domain mutation (**e**), Vaco481 CRC cells with a *PIK3CA* Q546P mutation, or a CRC PDX with a *PIK3CA* E542K mutation (**g**). **h** Lysates of *PIK3CA* E542K mutant PDX tumors treated with the indicated drug were blotted with the indicated antibodies. Data from 10 tumors/group are presented as mean +SEM. Statistical analyses, two-way ANOVA. The Alpelisib data presented in (**a**) and (**b**) are the same dataset. Source data are provided as a Source Data file.

a *PIK3CA* E542K mutation. As expected, compared to *PIK3CA* E542K mutant PDXs treated with vehicle control, Alpelisib reduced pAKT levels only, Tazemetostat reduced H3K27me3 only, whereas the drug combination decreased levels of both pAKT and H3K27me3 (Fig. 7h). It is worth noting the drug combination was well-tolerated as the body weights of mice were maintained during the course of the drug treatments (Figure S7a–g). Moreover, the drug combination inhibited tumor growth synergistically in CRCs with a *PIK3CA* helical domain mutation (Figure S7h), but had only an additive effect on CRCs with either WT *PIK3CA* or a *PIK3CA* H1047R mutation (Figure S7h). These results suggest that the combination of Tazemetostat and Alpelisib could be an effective treatment for cancers harboring *PIK3CA* helical domain mutations.

## Discussion

Our study reveals a previously unrecognized mechanism by which *PIK3CA* helical domain mutations exert oncogenic signaling: p85β, but not p85α, dissociates from the p110α helical domain mutant protein and translocates into the nucleus. The nuclear p85β stabilizes EZH1/2 by recruiting deubiquitinase USP7 to the two proteins and enhancing H3K27 trimethylation. Additionally, our previous study demonstrated that the p110α helical domain mutant proteins directly bind to IRS1 and activate the canonical PDK1-AKT signaling pathways[12]. Therefore, *PIK3CA* helical domain mutations promote oncogenesis through two independent pathways: a canonical p110-PDK1-AKT pathway and a nuclear p85β-USP7-EZH1/2 axis (Fig. 8). Moreover, our data suggest that targeting both pathways with Alpelisib and

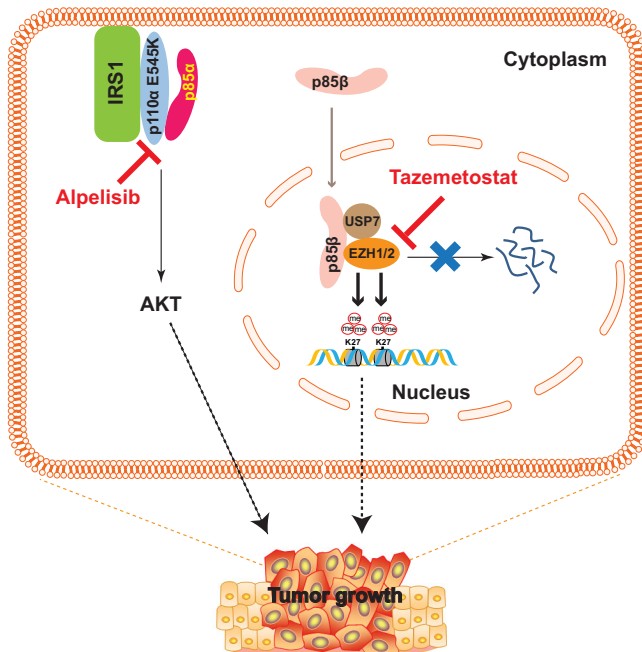

**Fig. 8 A model for how *PIK3CA* helical domain mutations promote oncogenesis.** *PIK3CA* helical domain mutations promote oncogenesis through two independent pathways: (1) p110α helical domain mutant protein directly interacts with IRS1 to activate the canonical PDK1-AKT pathway[12]; and (2) p85β translocates into the nucleus, stabilizes EZH1/2 by recruiting deubiquitinase USP7, and enhances H3K27 trimethylation. Simultaneously targeting both pathways with Alpelisib, a p110α inhibitor, and Tazemetostat, an EZH2 inhibitor, induces regression of tumors harboring a *PIK3CA* helical domain mutation.

Tazemetostat could be an effective therapeutic approach for *PIK3CA* helical domain mutant cancers.

Firstly, this study sheds new light on the nuclear translocation and function of p85β. We identified an NLS (ELQMKRTAIEAF) in p85β that plays a major role in its nuclear translocation. When we mutated the critical basic amino acids KR to AA in both ectopically expressed and endogenous p85β, the mutant p85β protein failed to translocate into the nucleus (Fig. 4 and S4). However, the p85β NLS is not sufficient to induce nuclear translocation, as our data showed that p85β translocates into the nucleus in the *PIK3CA* helical domain mutant cell lines, but not in the WT and *PIK3CA* kinase domain mutant cell lines (Fig. 3). We postulate that the interaction between IRS1 and p110 helical domain mutant protein results in the release of p85β from the PI3K complexes, which in turn exposes the NLS in the iSH2 domain of p85β and triggers its nuclear translocation. Although BRD7 has been reported to act as a chaperone for nuclear transport of p85α and p85β[30,31], our data suggest that BRD7 is not the major mediator of p85β nuclear translocation in *PIK3CA* helical domain mutant cancer cells, because knockout of BRD7 only had a marginal effect on nuclear p85β levels. It is interesting that p85β, but not p85α, dissociates from the p110α helical domain mutant proteins. Our domain-swapping experiment shows that the N-terminal p85β sequences cause its dissociation from the p110α helical domain mutant proteins (Fig. 1i, j). Although p85α also has a putative NLS sequence, it still tightly binds to p110α helical mutant protein, which prevents it from NLS-mediated nuclear translocation.

Secondly, our data suggest that the nuclear p85β plays an oncogenic role in tumors. Nuclear p85β has been shown to interact with XBP1 to modulate endoplasmic reticulum stress or binds to BRD7 and XBP1 to regulate glucose homeostasis[27–31,38]. Although it has been proposed that overexpression of p85β in some tumor types promotes cancer progression through the canonical PI3K enzymatic activity[22], none of the previous studies have implicated nuclear p85β in tumorigenesis. Here, we provide several lines of evidence implicating an oncogenic role of nuclear p85β in *PIK3CA* helical domain mutant cancers: (1) knockout of p85β reduces xenograft tumor growth of *PIK3CA* E545K mutant cells (DLD1 E545K-only and H460), but not the *PIK3CA* WT cells (DLD1 WT-only) and *PIK3CA* H1047R mutant cells (RKO); (2) knockdown of p85β reduces the growth of a panel of *PIK3CA* helical domain mutant cell lines, not a panel of *PIK3CA* kinase domain mutant cell lines; (3) the p85β NLS mutant DLD1 knockin cells, lacking nuclear translocation of p85β, have reduced xenograft tumor growth.

Thirdly, our data suggest that the nuclear p85β stabilizes EZH1 and EZH2 by recruiting deubiquitinase USP7, and enhances H3K27 trimethylation, thereby promoting the growth of *PIK3CA* helical domain mutant tumors. Consistently, an oncogenic role of EZH1/2, especially EZH2, has been well-documented because recurrent gain-of-function EZH2 mutations have been identified in 22% of diffuse large B-cell lymphomas and ~10% of follicular lymphomas[39]. Interestingly, the aforementioned oncogenic nuclear p85β function seems to be independent of p110α and p110β, because our data demonstrated that EZH1 and EZH2 bind to p85β, but not p110α and p110β (Fig. S6a). Although our data suggest that the nuclear p85β-USP7-EZH1/2 pathway is activated in cancer cells with a *PIK3CA* helical domain mutation, we cannot rule out the possibility that this pathway could also be triggered by certain physiological stimuli in normal cells as well.

Lastly, our data suggest that simultaneously targeting nuclear p85β-stabilized EZHs and p110α could be an effective cancer treatment. The p110α specific inhibitor Alpelisib in combination with Fulvestrant has been approved by the FDA for the treatment of HR-positive and HER2-negative breast cancers with *PIK3CA* mutation[40]. However, the efficacy of Alpelisib in other tumor types (such as colorectal cancer) has been disappointing[41]. Moreover, in some early clinical trials, patients with *PIK3CA* helical domain mutations are more resistant to Alpelisib than those with *PIK3CA* kinase domain mutations[16,42]. Our study provides a mechanistic explanation to this observation, because *PIK3CA* helical domain mutations promote tumorigenesis through both the canonical PI3K-AKT and the newly discovered nuclear p85β-USP7-EZH1/2 pathways. The data suggest that targeting both pathways could be an effective approach to treat *PIK3CA* helical domain mutant cancers. In fact, we have demonstrated that the combination of Alplelisb and EZH2 inhibitor Tazmetostat induced regression of tumors harboring each of the three recurrent *PIK3CA* helical domain mutations, but not tumors with *PIK3CA* WT or a kinase domain mutation. It is worth noting that Cha and colleague reported that AKT phosphorylates EZH2 and regulates its methyltransferase activity[43]. Thus the drug combination may impact EZH2 through two different pathways. Currently, we have only tested the drug combination in CRC models. However, the molecular mechanisms we uncovered here apply to other types of tumor types with a *PIK3CA* helical domain mutation as well. Ultimately, the efficacy of the drug combination needs to be tested in cancer patients. We are actively pursuing a phase I clinical trial of the combination Alpelisib and Tazemetostat in patients whose tumors harbor a *PIK3CA* helical domain mutation.

## Methods

**Reagents.** Chemicals, antibodies, and other reagents are listed in Table S2.

**Tissue Culture and transfection.** Colorectal cancer (CRC) cell lines DLD1, HCT116, RKO, SW480, LoVo, SW948, H460, T47D, HEK 293 T were purchased

from the ATCC. Colorectal cancer (CRC) cell lines DLD1, HCT116, RKO, SW480, LoVo, SW948and genetically engineered isogenic cell lines DLD1 *PIK3CA* E545K cells and DLD1 *PIK3CA* WT cells were grown in McCoy's 5 A medium (Gibco) supplemented with 10% of fetal bovine serum (Gibco). Lung cancer cell line H460 and breast cancer cell line T47D were cultured in RPMI 1640 medium (Sigma) containing 10% of FBS. Breast cancer cell line MDA-MB361 was maintained in Leibovitz's L-15 medium (Gibco) with 20% of FBS. Human embryonic kidney HEK 293 T cells were cultured in DMEM medium (Sigma) containing 10% FBS. Penicillin/Streptomycin (1%) was added to tissue culture media for all cultures. Cells were incubated at 37 °C in a humidified atmosphere with 5% CO$_2$. All cell lines were tested routinely to avoid *Mycoplasma* contamination (Yeasen, cat # 40601ES20). The cell lines were authenticated by the Genetica DNA Laboratories using STR profiling. Transfection was conducted using Lipofectamine 3000 reagent (Life Technologies) according to the manufacturer's instructions.

**DNA constructs and mutagenesis**. The plasmids we constructed in this study are listed in Table S2. The primers which were used for vector construction are listed in Table S3. Briefly, pCMV backbones (Invitrogen) were used for gene expression in mammalian cells using the USER cloning system[44]. LentiCRISPR V2 backbone (Addgene) was used for gene knockout in cells. Point mutations in constructs were generated using a Site-Directed Mutagenesis Kit (Agilent). pAAV-loxP-Neo vector was used for homologous recombination of endogenous p85β NLS point mutation.

**CRISPR/CAS9 genome editing**. Three different guiding RNA pairs for p85βknockout were designed using the IDT design tool (https://sg.idtdna.com/pages) and cloned individually into the lentiCRISPRv2 vector as described previously[45]. DLD1 isogenic cell lines with *PIK3CA* E545K or with wild-type *PIK3CA* were transfected with these vectors. After 48 h, cells were trypsinized, and stable clones were selected using 1.5 μg/ml puromycin (Invitrogen) for 2 weeks. Knock-out of p85β was screened using genomic PCR and validated by Western blot.

For CRISPR/CAS9 mediated NLS point mutation on endogenous *PIK3R2* locus, 3 different guide RNA pairs surrounding NLS mutation sites of *PIK3R2* locus were designed and cloned. Homologous arms of the NLS target sites were mutated and cloned into the pAAV-loxP-Neo vector. Targeting vectors were co-transfected with individual gRNA vectors into DLD1 cells. p85β NLS mutated cell clones were screened by genomic PCR and verified by genomic DNA sequencing.

**siRNA knockdown**. The siRNAs targeting human *PIK3R2*/p85β and the scramble siRNA control were purchased from Biotend (Shanghai, China). siRNAs were performed as described previously[36]. Cells were harvested 48–72 h post-transfection for various assays.

**RNA extraction and Quantitative Real-Time PCR**. Total RNA was extracted and purified using TRIzol (Invitrogen) according to the manufacturer's instructions, and 1 μg of total RNA was reverse transcribed using the PrimeScript RT Reagent Kit (TaKaRa, Japan). The gene expression levels were measured by a quantitative real-time PCR system (Qiagen, Germany). β-tubulin was used as the reference gene for normalization. The qRT-PCR primers are listed in Table S3.

**Cell growth assays**. For cell proliferation, 3000 cells per well were seeded in a 96-well plate. Cell viability was measured for five consecutive days using Cell Counting Kit-8 (Dojindo, Japan) according to the manufacturer's instructions. Absorbance at OD450 was used to plot cell growth curves. For clone formation assay, the same number of cells were seeded in 6-well plates and maintained in McCoy's 5 A medium with 1% FBS. After 14 days, cells were washed with PBS and stained with 0.5% crystal violet.

**Immunofluorescence staining**. Immunofluorescence staining was performed as described previously[46]. Briefly, cells were seeded on coverslips in a 6-well plate. After 24 to 48 h, cells were fixed in 4% paraformaldehyde for 20 min, permeabilized with 0.5% Triton X-100 for 30 min, and blocked in 10% goat serum for 1 h at room temperature. The cells were then incubated with p85β antibodies in 10% goat serum at 4°C overnight, followed by incubation with secondary fluorochrome-labeled antibodies for 40 min at 37 °C. After incubation with DAPI for 3–5 min at room temperature to stain the nucleus, cells were washed three times with PBS and imaged with a confocal laser scanning microscope.

**Co-immunoprecipitation and Western blotting**. Co-immunoprecipitation (Co-IP) was performed as previously described[47]. For transfection-based Co-IP assays, cells were transfected with indicated vectors and incubated in 1 mL of lysis buffer (50 mM Tris-HCl at pH 7.5, 1 mM EDTA at pH 8.0, 150 mM NaCl, 1% NP-40, cOmplete Protease Inhibitor, PhosSTOP, and PMSF). Cell lysates were immuno-precipitated with indicated primary antibodies overnight at 4 °C and then Protein A/G for 2 h. The beads were washed three times with the lysis buffer and eluted in SDS sample buffer. The eluted immunocomplexes were resolved by SDS-PAGE, followed by Western blotting.

**Nuclear/cytoplasmic fractionation**. Cell pellets were resuspended in 1 ml fractionation buffer (0.1% NP-40, cOmplete Protease Inhibitor, PhosSTOP, and PMSF in PBS buffer) and gently pipetted up and down 15 times and then centrifuged at 13,500 g for 30 s immediately. The supernatants were labeled as cytoplasmic fractions. The pellets were washed twice with fractionation buffer and then dissolved in 160 μl of fractionation buffer as the nuclear fraction. Each fraction was sonicated 10 s at 60% output settings.

**Xenografts**. All animal experiments were performed in accordance with protocols approved by the IACUC committee at Case Western Reserve University. Xeno-grafts were established as described previously[48]. Briefly, for cells, three million cells were injected subcutaneously and bilaterally into athymic nude mice. For PDXs, two pieces of xenograft tumors (~2–4 mm$^3$) were inserted subcutaneously and bilaterally into athymic nude mice. Tumor volume was measured at the indicated time points and calculated as length × width$^2$/2.

**Drug treatment**. Alpelisib (BYL719) and Tazematostat (EPZ-6438) were dissolved in 0.5% carboxymethylcellulose sodium salt (CMC). GSK2816126 was dissolved in 20% Captisol. Once tumor sizes reached 100–150 mm$^3$, mice were randomly assigned into different groups (5 mice per group) and treated with vehicle, Alpelisib (BYL719, 12.5 mg/kg, oral gavage, once daily), EPZ-6438 [500 mg/kg, oral gavage, bid as described in[49]], GSK2816126 (25 mg/kg, once daily, I.P.), a combination of BYL719 and EPZ-6438, or a combination of BYL719 and GSK2816126.

**RNA-sequencing**. High-Throughput RNA sequencing was performed in DLD1 parental cell and DLD1 p85β NLS$^{KR-AA}$ mutant cell using standard procedures. Briefly, total RNAs were extracted using TRIzol reagents, and Illumina high-throughput sequencing libraries were constructed and sequenced according to the manufacturer's instructions. The raw paired-end reads were trimmed and quality controlled by SeqPrep (https://github.com/jstjohn/SeqPrep) and Sickle (https://github.com/najoshi/sickle) with default parameters. To identify differential expression genes (DEGs) between the two samples, the expression level of each transcript was calculated according to the fragments per kilobase of exon per million mapped reads (FPKM) method. Differential expression of genes (DLD1 vs. p85$^{KR-AA}$ Mut clones, fold change ≥3 or ≤−3 and adjusted p-value < 0.01) were analyzed and selected for subsequent analysis.

**ChIP-seq**. Chromatin Immunoprecipitation (ChIP) was performed using the SimpleChIP Enzymatic Chromatin IP Kit (Cell Signaling Technology) following the manufacturer's instructions. Chromatin from DLD1 and DLD1 p85β NLS$^{KR-AA}$ mutant cells were sonicated and immunoprecipitated with an anti-H3K27me3 antibody. Purified ChIP DNA was verified on an agarose gel to ensure proper fragmentation for Illumina library construction. Next-gene sequencing was performed using an Illumina Hi-Seq 2000 machine. Raw reads (single-end, 50 bp) were aligned to human reference genome hg19 using bowtie (v1.1.0), allowing up to one mismatch. Peak calling was performed using model-based analysis of ChIP–seq (MACS, v1.4.2) with a cutoff of $P \leq 1 \times 10^{-8}$. Promoter regions were defined as regions 3 kb upstream to 3 kb downstream of the TSS. The H3K27me3 occupancies at promoter regions were normalized as reads per kilobase per million reads (RPKMs). Fold changes and significance of H3K27me3 differences between conditions were determined using an MA-plot-based method with a random sampling model, which was implemented in the R package DEGseq. Enriched regions were determined by the HOMER program (http://homer.salk.edu/homerl) with an FDR value cutoff of 0.05. ChIP-seq density heatmaps and histograms were generated using ngs.plot. The validation experiment was performed using ChIP-PCR with primers listed in Table S3.

**Immunohistochemistry**. Immunohistochemistry was performed as described previously[50,51]. Briefly, paraffin-embedded mouse and human tissues were deparaffinized in xylene and antigen retrieved by boiling the sample for 20 min. Samples were incubated with primary antibodies at 4 °C overnight. The sections were stained with secondary antibodies for 30 min at room temperature and then stained with an EnVision-HRP kit (Dako).

**Mining the TCGA datasets**. The dataset files of colorectal cancer (COAD), bladder carcinoma (BLCA), endometrial carcinoma (UCEC), and breast cancer (BRCA) were downloaded from the TCGA website. The files include the RNA-seq files providing normalized FPKM values, the somatic mutations, and the 5-year survival of each patient.

For gene expression analysis, FPKM values of indicated genes from tumor samples and corresponding normal tissue samples, if available, were plotted. The statistical significance difference of tumor versus non-tumor was calculated using the student t-test.

To analyze the association of *PIK3R2* expression with 5-year survival, patients were divided into three groups according to their *PIK3CA* mutation status: helical domain mutation group (Patients with *PIK3CA* mutation at E542, E545, and Q546), non-helical domain mutation group (Patients with *PIK3CA* mutation at H1047 and other sites) and wild type group (Patients with wild-type *PIK3CA*), and then

patients were further divided into *PIK3R2* high and *PIK3R2* low according to the median expression of *PIK3R2* in each group. Due to the limitation of the patient number of helical domain mutation groups in individual tumor type, *PIK3R2* high and *PIK3R2* low in each group of four tumor types were combined to assess the relevance of *PIK3R2* expression and 5-year survival of all patients. Kaplan-Meier analysis of 5-year survival was performed with a Cox proportional hazards model.

**Quantification and statistical analysis**. GraphPad Prism software was used to create the graphs. Data are plotted as mean ± SEM. We applied the two-sided *t*-test to compare the means between the two groups, assuming unequal variances. For xenograft growth, we carried out ANOVA for repeated measurements to test whether there is an overall difference in the tumor sizes by testing group differences as well as whether there was a difference in the development of tumor sizes over time between the two groups by testing the interaction between time and group. The combinatorial effect of drug treatment in vivo was evaluated by combination ratio using the fractional product method. FTV (fractional tumor volume) = mean of final tumor volume of treated group/mean of final tumor volume of the control group. Expected FTV = (FTV of agent1) * (FTV of agent2). Observed FTV = mean final tumor volume of combinatorial treatment/means final tumor volume of the control group. Combination ratio = expected FTV/observed FTV. Combination ration > 1, = 1, or < 1, was defined as synergy, additive effect, or antagonism, respectively.

**Reporting summary**. Further information on research design is available in the Nature Research Reporting Summary linked to this article.

## Data availability

Source data are provided with this paper as an excel file and original Western Blot images are provided in Figure S8. The RNA-seq and ChIP-seq data are deposited in the NCBI GEO database and the accession numbers are GSE190740 and GSE190434. Source data are provided with this paper.

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

## Acknowledgements

This work was supported by NIH grants R01CA264320, R01CA196643, P50CA150964, and a Stand Up to Cancer Colorectal Cancer Dream Team Translational Research Grant (SU2C-AACR-DT22-17) to Z. Wang. Stand Up to Cancer is a program of the Entertainment Industry Foundation. Research grants are administered by the American Association for Cancer Research, a scientific partner of SU2C. This work was also supported by the Program for Professor of Special Appointment (Eastern Scholar) at Shanghai Institutions of Higher Learning (TP2017027); the Program of Shanghai Academic/Technology Research Leader (19XD1423500); Shanghai Municipal Education Commission-Gaofeng Clinical Medicine Grant Support (20171915); National Natural Science Foundation of China (81772503, 82073044); and State Key Laboratory of Oncogenes and Related Gene (91-17-10, ZZ2016SYL) to Y. Hao. This research was further supported by the Gene Expression and Genotyping Facility of the Case Comprehensive Cancer Center (P30 CA043703) directed by M. Veigl.

## Author contributions

Z.W. and Y.H. conceived the experiments. Y.H., B.H., L.W., Y.L., C.W., T.W., Y. Zhang, L.S., Y. Zhan, Y. Zhao, and M.V. performed the experiments and analyzed the data. S.M. provided essential reagents. Z.W., Y.H., R.C. and B.H. wrote the manuscript.

## Competing interests

The authors declare no competing interests.
