## [Peer Review File · Nature Communications]

REVIEWER COMMENTS

Reviewer #1 (Remarks to the Author):

In the present manuscript, Hao et al describe a mechanism through which the class-I PI3K regulatory subunit p85 β dissociates from mutated p110 α and translocate to the nucleus. They propose that nuclear p85 β recruits the deubiquitinase USP7 to stabilize the polycomb group protein, EZH1 and EZH2, resulting in increased H3K27 trimethylation. Accordingly, by interfering with p85 β nuclear relocalization with either knockdown or nuclear localization signal mutations, the authors reported reduced growth of tumours harbouring PIK3CA helical-domain mutations.

They propose that the oncogenic function of E545K PIK3CA mutation depends on the release of p85 β from mutated p110 α , its subsequent translocation to the nucleus and the consequent increased activity of EZH proteins. In support to their findings, treatment of xenografts tumors harbouring PIK3CA helical-domain mutations with a combination of a p110 α inhibitor Alpelisib and an EZH inhibitor (Tazemetostat) leads to tumor regression.

Understanding how mutations in PIK3CA exert their oncogenic role is an important and poorly understood question in the cancer field. The model presented here is interesting and, overall, is supported by convincing results. However, there are some important issues to resolve.

The following points need to be addressed to strengthen the manuscript

Major points:

1. The authors propose that p85 β translocate to the nucleus in the presence of mutations in the helical-domain of PIK3CA. Is this mechanism effective in normal cells under physiological conditions? If not, how and why a selective pressure in cancer cells should be responsible for the selection of this molecular mechanism not observed in normal conditions?
2. The authors report that combination treatment with Alpelisib and EZH inhibitor is effective in PIK3CA helical domain mutant tumors (E545K). Nevertheless, Arteaga group (Clin Can Rese 2017) reported that mutations on exon 20 (H1047R) appear to be associated with increased clinical durable responses, in contrast to exon 9 mutations (E545K). In addition, while Alpelisib demonstrated clinical activity in breast cancer patients with PIK3CA mutations, in mCRC PIK3CA mutations are associated with intrinsic and acquired resistance (Bernards and Opdam, BJC, 2021). The authors need to reconcile their findings with current literature.
3. Experiments are only performed on a single line of colorectal cancer (DLD1) w/o E545K mutation and some breast cancer cell lines. Can the author demonstrate that the proposed mechanism is conserved in other cellular models/tumors harbouring similar PIK3CA mutations?
4. A live-cell imaging experiment in cells expressing either fluorescently-tagged wild type or E545K p110 α together with fluorescently-tagged p85 β could be informative to directly visualize p85 β relocalization to the nucleus in the absence of interaction with p110 α .

5. In figure 4a, the authors identified a putative nuclear localization sequence (NLS) in p85 β . They showed that two point mutations introduced in wild type p85 β are sufficient to abolish p85 β translocation to the nucleus. However, the introduced mutations might have changed the p85 β proper folding thus resulting in its loss from the nucleus. Cloning of the 474-484 amino acids sequence fused with a fluorescent-tag would be more informative to visualize its intracellular localization and to support the conclusion that the predicted NLS in p85 β is actually responsible for its relocation to the nucleus.

Minor points:

1. In figure 1g, h in DLD1 cells, why heterozygous cells show less binding than KO cells? Moreover, always referred to DLD1, the wild type p110 α control is missing.

2. In figure 3a, p85 β appears localized mostly outside or in close proximity to the nucleus. Immunofluorescence staining with anti-Laminin B can be informative to better understand p85 β localization.

3. In figure 5b, EZH seems stabilized also in H1047R. How can this be explained?

4. In figure 5e, p85 α should be added as a control.

5. At page 9, end of first paragraph, the sentence “Those data suggest that p85 α , but not p85 β , is the major regulatory subunit for PI3K activity in DLD1 colorectal cancer cells” is an overstatement.

Reviewer #2 (Remarks to the Author):

Summary:

While existing evidence demonstrate that PIK3CA catalytic and helical domain mutations exert oncogenicity through distinct molecular mechanisms, the details surrounding the different mechanisms utilized by proteins encoding these common genomic alterations is poorly understood. Research focused in this area has the potential to define novel therapeutic combinations, as well as specific genomic contexts in which therapies are likely to produce a durable and efficacious outcome for patients. To this purpose, the authors demonstrate that the PI3K regulatory protein p85-beta dissociates from the helical domain mutant p110a proteins, supporting p85b translocation into the nucleus. Using a combination of isogenic knockout and/or genetic mutant cell lines and multiple cancer cell models, the authors suggest that nuclear p85b leads to the deubiquitination and stabilization of EZH1/2, thereby increasing H3K27me3. Moreover, in vivo experiments demonstrate that cancers characterized by PIK3CA helical domain mutations are sensitive to the combined inhibition of p110a and EZH1/2. The authors utilize a combination of mechanistic studies and patient derived xenograft models to test the hypothesis that combined EZH and PI3K inhibition results in tumor regression in tumors characterized by PIK3CA helical domain mutation. Collectively, these data provide evidence that the combination of alpelisib an tazemetostat may be a viable therapeutic regimen for genomically defined PIK3CA helical domain

mutated cancers such as those containing a PIK3CA E542K or E545K mutation, which is inherently a novel and potentially clinically actionable finding.

Many of the experiments presented were genetically elegant, such as using the chimeric proteins to define the region of p85b which supports dissociation with p110a helical domain mutations, and mutating the p85b NLS. Other experiments were thorough, such as using a combination of PIK3CA helical mutant cell lines and PDXs for the in vivo inhibitor experiments. Combining these features into a curated dataset that includes the utilization of mechanistic studies, patient samples, and PDX models to test their hypothesis is a clear strength.

However, enthusiasm is reduced as some of the experiments leave out PIK3CAH1047R as a specific experimental condition to rigorously test their hypothesis that this property is unique to tumors characterized by helical domain PIK3CA mutations. In order to prove that helical domain PIK3CA mutations uniquely dissociate p85b to contribute to tumor oncogenicity, then experiments at hand would benefit from the inclusion of PIK3CAH1047R as a second negative control alongside wildtype PIK3CA. Further, some experiments do not utilize the appropriate cell lines (e.g., PIK3CA wild-type DLD cells are used as opposed to PIK3CA E545K DLD cells, or better, including both), which lacks the rigor necessary to define the nuclear p85b-dependent molecular mechanism. Lastly, a more robust inclusion of and detail associated with the differential gene expression and how modulating EZH1/2 expression and subsequently H3K27me3 (and gene expression via this mechanism) may be involved in this molecular mechanism would greatly improve and complete this manuscript.

Major points:

1. While generally the experiments presented are well-controlled, a major point the manuscript addresses is the unique mechanism by which helical domain mutations (and not the catalytic domain mutation H1047R) dissociate from p85b to contribute to tumorigenicity. As such, the manuscript could be improved by inclusion of a third condition in many experiments: (1) wild type; (2) PIK3CA E542/E545K; and (3) PIK3CA H1047R. Examples of where this additional condition would benefit the proposed mechanism is in experiments where the isogenic DLD cells are used, amongst others: Figures 2b, 2e/f, 3a. This is especially important as the heterogeneity and other confounding genomic alterations present in the cancer cell lines utilized throughout the manuscript (e.g, RKO, T47D, etc), make a linear comparison challenging.
2. It is unclear whether the p85b NLS mutant knock-in cells were made in the DLD background that expresses a helical domain PIK3CA mutation. Considering only helical PIK3CA mutants rely on nuclear p85b to support oncogenicity, it seems essential that the knockin cells should be made in the helical domain background. If they were not, this could explain the relatively low number of genes that have increased expression in the NLS knockin cells. (please see below points)
3. In Figures 5A-C, a reduction in H3K27me3 following either p85b NLS mutant expression or p85b knockout is visible at the global scale by using western blot. By eye it appears that the loss of nuclear p85b may reduce H3K27me3 by as much as 75% (see Fig 5b, MB-361 cells for the greatest magnitude reduction in H3K27me3). Based on these results, one would hypothesize that the ability of nuclear p85b to regulate H3K27me3 would result in the widespread loss of promoter H3K27me3 and widespread corresponding increase in gene expression. However, only 137 genes exhibited an increase in gene expression using the isogenic p85b NLS mutant cell lines, which is inconsistent with the western data and suggests additional mechanism(s) may be play here. In order to rigorously test this hypothesis, H3K27me3 ChIP-seq should be performed to confirm that the differential gene expression is a result of

genomic loci-specific changes to H3K27me3. Lastly, the logic as to why H3K27me3 was selected and followed up on is unclear, especially since AKT has already been shown to regulate H3K27me3 via EZH2 S21 phosphorylation (Cha, et al Science 2005).

4. Similar to the above point, why does loss of p85b dramatically reduce both EZH1 and EZH2, but only influence the expression of 253 genes? EZH1/2 are the main H3K27 methyltransferases, with only the K3K9 methyltransferase G9a able to methylate H3K27 under select circumstances. Thus, one would hypothesize a greater impact on gene expression if EZH1/2 expression is destabilized.

5. Linking to points 3-4 above, while it may be difficult to justify how such dramatic changes to EZH1/EZH2 expression and H3K27me3 results in limited changes to gene expression, the manuscript could be strengthened by inclusion of the RNA-seq data in a main figure, and including the identity of the differentially regulated genes. This is especially true if key oncogenes/tumor suppressors are differentially regulated. This could be even further enhanced if the differentially regulated genes could be linked to changes to genomic loci-specific promoter H3K27me3.

6. The cells utilized in Fig 5F appear to be wild type PIK3CA. However, as shown in Fig 3, only cells that express PIK3CA helical mutations are characterized by high p85b nuclear levels. Thus, if nuclear p85b is required for this mechanism described in this paper, it is essential that the cells utilized in experiments such as Fig 5F contain a helical PIK3CA mutation in the presence and absence of USP7 KO. Currently, the experiment shown in Fig 5F demonstrates that USP7 regulates EZH1/2 stability independently from PIK3CA helical domain mutant expression and may not fundamentally be different than what has been previously published (Lee J, et al, and Zheng, N, et al. Refs 32 and 33). This comment is also viable for Fig 5g and h.

7. The discussion does not include include commentary on the existing literature that demonstrates EZH2 is an AKT substrate (Cha, et al Science 2005), and thus inhibition of EZH1/2 may support PI3K inhibition for more than just the proposed mechanism via USP7 stabilization.

8. Synergy calculations (Bliss/Loewe) could be performed to enhance the value of combination therapy in helical domain mutant cell lines and tumors, in comparison to PIK3CA wild-type and PIK3CA catalytic domain mutants.

Minor Points:

1. Accessory experiments demonstrating the ability of E542/E545K to shift p85b into the nucleus could be performed with H1047R to solidify the proposed differences in molecular mechanisms by which PI3K mutations confer oncogenicity. E.g, overexpress H1047R in cells and evaluate p85b subcellular localization in a manner analogous to what was done for E545K in Fig S3E.

2. Inclusion of IHC for subcellular localization of p85b in H1047R-mutated colon cancer specimens as part of Fig S3F would be beneficial

3. Why is it stated that global gene expression is being regulated if a total of 253 genes are differentially regulated in the p85bKR-AA cells?

4. Text in the results section describing Fig 5A states, "... We thus examined histone modifications..." However, only H3K27me3 was evaluated in Fig 5. Fig 5 would benefit from inclusion of Fig S5b in the main figure 5.

5. As an extension of Figure 5, the conclusion would be strengthened by using DLD cells that express PIK3CA E545K with or without USP7 KO, and then evaluating whether H3K27me3 is preferentially reduced in the E545K + USP7 KO cells.

6. Tazemetostat is spelled incorrectly in the last sentence prior to the discussion

7. The very end of the discussion is redundant; describes the FDA approval of alpelisib and associated information twice in the same paragraph.

** See Nature Research's author and referees' website at www.nature.com/authors for information about policies, services and author benefits.

Responses to the reviewers' comments

We thank the reviewers for their high enthusiasm for our manuscript. We also appreciate their constructive criticisms, responding to which help to improve the manuscript. New additions/changes are highlighted in red in the revised manuscript. Below is our point-by-point response to their concerns:

Responses to the comments of reviewer 1.

Major points:

1. **“The authors propose that p85 β translocate to the nucleus in the presence of mutations in the helical-domain of PIK3CA. Is this mechanism effective in normal cells under physiological conditions? If not, how and why a selective pressure in cancer cells should be responsible for the selection of this molecular mechanism not observed in normal conditions?”**

Response: We reported previously that PIK3CA/p110 α helical domain mutant proteins gain a direct interaction with IRS1 independent of p85, which in turn brings the PI3K complexes to the membrane to access its substrate PIP2 and activates the downstream signaling independent of a growth factor stimulation (Hao Y et al, *Cancer Cell* 23:583 2013). In this manuscript, we reveal that p85 β dissociates from the PI3K complexes with a p110 α helical domain mutation specifically. We postulate that the dissociation of p85 β from the helical domain mutant p110 α may be due to the binding of helical domain mutant p110 α to IRS1, although the detailed mechanism remained to be determined. We have not found this mechanism occurring in normal cells under physiological conditions. We have now incorporated these concepts into the discussion section.

2. **“The authors report that combination treatment with Alpelisib and EZH inhibitor is effective in PIK3CA helical domain mutant tumors (E545K). Nevertheless, Arteaga group (Clin Can Rese 2017) reported that mutations on exon 20 (H1047R) appear to be associated with increased clinical durable responses, in contrast to exon 9 mutations (E545K). In addition, while Alpelisib demonstrated clinical activity in breast cancer patients with PIK3CA mutations, in mCRC PIK3CA mutations are associated with intrinsic and acquired resistance (Bernards and Opdam, BJC, 2021). The authors need to reconcile their findings with current literature.”**

Response: Those data are consistent with our findings. First, Arteaga's study shows that cancer patients whose tumors harbor a PIK3CA helical domain mutations are resistant to a PI3K inhibitor compared to patients whose tumors harbor a PIK3CA kinase domain mutation. Our study provides a mechanistic explanation for this observation. While the PIK3CA kinase domain mutation activates the canonical PI3K-AKT pathway, the PIK3CA helical domain mutations activate the nuclear p85 β -EZH pathway to promote tumorigenesis in addition to the canonical PI3K-AKT pathway. Therefore, targeting the p110 α activity alone is not sufficient for tumors with a PIK3CA helical domain mutation. The Bernards and Opdam study affirms this concept, as their data shown in Table S2 of their paper indicate that the three non-responders to the encorafenib + cetuximab + alpelisib (triple treatment) whose tumors with PIK3CA mutations are all helical domain mutations (E542K, E545K, and Q546P). In contrast, the partial responder

patient whose tumor with a PIK3CA mutation is the H1047R kinase domain mutations. We have now incorporated these discussions in the revised manuscript.

3. “Experiments are only performed on a single line of colorectal cancer (DLD1) w/o E545K mutation and some breast cancer cell lines. Can the author demonstrate that the proposed mechanism is conserved in other cellular models/tumors harbouring similar PIK3CA mutations?”

Response: We have included more cell lines to demonstrate the generality. Specifically:

- 1) In the original manuscript, we showed that p85 β dissociated from mutant p110 α two cell lines with a PIK3CA E545K mutation (DLD1 and H460, a lung cancer cell line), but not in two cell lines with a PIK3CA kinase domain mutation (HCT116 and T47D) or in two cell lines with WT PIK3CA (SW480 and LoVo). We have now performed the experiment with three additional cell lines: MB361 with a PIK3CA E545K mutation, SW948 with a PIK3CA E542K mutation, and RKO with a PIK3CA H1047R kinase domain mutation (Fig. S1i).
- 2) We now knocked out p85 β in two additional cell lines, H460 and RKO, and demonstrated that knockout p85 β in H460, which harbor a PIK3CA E545K mutation, reduced cell growth, colony formation, and xenograft tumor growth (new Fig. 2 f to i), whereas knockout of p85 β in RKO, which harbor a PIK3CA kinase domain mutation, had no impact on cell growth, colony formation and xenograft tumor growth (new Fig. 2 f to i). Additionally, we showed that knockdown of p85 β reduced cell growth and colony formation of MB-361 and SW948 with a PIK3CA E545K and E542K respectively, but not in two cell lines SW480 and T47D with WT PIK3CA and a PIK3CA kinase domain mutation, respectively (Fig. S2 b to d).
- 3) In the original manuscript, we performed immunofluorescent staining of p85 β in eight different cell lines: 3 with a PIK3CA E545K mutation (DLD1, H460, and MB-361), 3 with a PIK3CA H1047R kinase domain mutation (HCT116, RKO, and T47D), and two with WT PIK3CA (SW480 and LoVo). We now performed immunofluorescent staining and cell fractionation of p85 β in SW948 cells, which harbor a PIK3CA E542K mutation, and the results show that p85 β translocates into the nucleus (Fig. 3 d & e).
- 4) Due to technical limitations, we only perform p85 β NLS mutant knockin in DLD1 cells because DLD1 is a near-diploid cell line that makes it feasible to mutate both alleles. In contrast, the other PIK3CA helical domain mutant cell lines are aneuploidy, which is very challenging to mutate the many alleles of PIK3R2/p85 β .
- 5) We now have data showing that knockdown of p85 β reduces the levels of H3K27me3, EZH1, and EZH2 in three additional PIK3CA helical domain mutant cell lines H460, MB-361, and SW948 (Figure 5d). Moreover, we also provide in the revised manuscript additional data showing that knockdown of p85 β does not impact the levels of H3K27me3, EZH1, and EZH2 in PIK3CA WT (SW480) and PIK3CA H1047R mutant cell lines (RKO, HCT116, and T47D; Figure 5e).
- 6) We have new data showing that p85 β associates with USP7, EZH1, and EZH2 in two additional PIK3CA E545K mutant cell lines MB-361 and H460 (Figure 6b and Figure S6b), but not in three PIK3CA H1047R cell lines HCT116, RKO, and T47D and a PIK3CA WT cell line SW480 (Figure 6b and Figure S6b). Moreover, we now have additional data showing that knockdown of p85 β in PIK3CA E545K mutant MB-361 cells reduces the binding of USP7 to EZH1 and EZH2 (Figure 6d).

- 4. “A live-cell imaging experiment in cells expressing either fluorescently-tagged wild type or E545K p110 α together with fluorescently-tagged p85 β could be informative to directly visualize p85 β relocalization to the nucleus in the absence of interaction with p110 α .”**

Response: We have now performed these experiments. The representative images are shown in Fig. S3h, and the videos are shown in supplementary materials videos 1 to 3.

- 5. “In figure 4a, the authors identified a putative nuclear localization sequence (NLS) in p85 β . They showed that two point mutations introduced in wild type p85 β are sufficient to abolish p85 β translocation to the nucleus. However, the introduced mutations might have changed the p85 β proper folding thus resulting in its loss from the nucleus. Cloning of the 474-484 amino acids sequence fused with a fluorescent-tag would be more informative to visualize its intracellular localization and to support the conclusion that the predicted NLS in p85 β is actually responsible for its relocation to the nucleus.”**

Response: We have now performed the suggested experiments (Fig. S4a). The results showed that the nuclear localization sequences (NLS) of p85 β made GFP solely localized in the nuclei, whereas KR-AA NLS mutant GFP fusion proteins were localized in both cytoplasm and nucleus just as the GFP protein alone.

Minor points:

- 1. “In figure 1g, h in DLD1 cells, why heterozygous cells show less binding than KO cells? Moreover, always referred to DLD1, the wild type p110 α control is missing.”**

Response: We’d like to apologize if any of our descriptions caused the confusion, but we do not have any KO cells shown in Fig. 1 g & h. In fact, both SW480 and LoVo cell lines harbor WT PIK3CA, which serves as WT p110 α controls.

- 2. “In figure 3a, p85 β appears localized mostly outside or in close proximity to the nucleus. Immunofluorescence staining with anti-Laminin B can be informative to better understand p85 β localization.”**

Response: We have now performed the suggested experiment (new Fig S3C). With the IF staining of laminin B, the results clearly show that p85 β is localized inside of the nucleus in PIK3CA E545K mutant cells, but not in PIK3CA WT-only cells.

- 3. “In figure 5b, EZH seems stabilized also in H1047R. How can this be explained?”**

Response: Sorry for the confusion. We have now performed experiments with additional cell lines, and the results clearly showed that knockdown of p85 reduces the levels of H3K27me₃, EZH1, and EZH2 in three additional PIK3CA helical domain mutant cell lines DLD1, H460, MB-361, and SW948 (new Figure 5d). Moreover, we also provide in the revised manuscript additional data showing that knockdown of p85 β does not impact the levels of H3K27me₃, EZH1, and EZH2 in PIK3CA H1047R mutant cell lines (RKO, HCT116, and T47D and in PIK3CA WT SW480 cells (new Figure 5e).

- 4. “In figure 5e, p85 α should be added as a control.”**

Response: We have now performed the experiment and showed that knockdown of p85 α did not affect the binding of EZH1 and EZH2 to USP7 (new Fig. 6e).

5. **“At page 9, end of first paragraph, the sentence “Those data suggest that p85 α , but not p85 β , is the major regulatory subunit for PI3K activity in DLD1 colorectal cancer cells” is an overstatement.”**

Response: We have now tuned down that statement as “Those data suggest that p110 proteins stabilized by p85 α in DLD1 cells.”

Responses to the comments of reviewer 2.

Major points:

1. **“While generally the experiments presented are well-controlled, a major point the manuscript addresses is the unique mechanism by which helical domain mutations (and not the catalytic domain mutation H1047R) dissociate from p85b to contribute to tumorigenicity. As such, the manuscript could be improved by inclusion of a third condition in many experiments: (1) wild type; (2) PIK3CA E542/E545K; and (3) PIK3CA H1047R. Examples of where this additional condition would benefit the proposed mechanism is in experiments where the isogenic DLD cells are used, amongst others: Figures 2b, 2e/f, 3a. This is especially important as the heterogeneity and other confounding genomic alterations present in the cancer cell lines utilized throughout the manuscript (e.g, RKO, T47D, etc), make a linear comparison challenging.”**

Response: As described in detail in our response to the major comment 3 of reviewer 1, we have performed additional experiments with a panel of cell lines as suggested here. The known gene mutations in these cell lines are listed in Figure S2h.

2. **“It is unclear whether the p85b NLS mutant knock-in cells were made in the DLD background that expresses a helical domain PIK3CA mutation. Considering only helical PIK3CA mutants rely on nuclear p85b to support oncogenicity, it seems essential that the knockin cells should be made in the helical domain background. If they were not, this could explain the relatively low number of genes that have increased expression in the NLS knockin cells. (please see below points).”**

Response: Yes, the p85 β NLS mutant knockin cells were made in parental DLD1 cells that are heterozygous for PIK3CA E545K mutation. We have now made this point clear in the revised manuscript.

3. **“In Figures 5A-C, a reduction in H3K27me3 following either p85b NLS mutant expression or p85b knockout is visible at the global scale by using western blot. By eye it appears that the loss of nuclear p85b may reduce H3K27me3 by as much as 75% (see Fig 5b, MB-361 cells for the greatest magnitude reduction in H3K27me3). Based on these results, one would hypothesize that the ability of nuclear p85b to regulate H3K27me3 would result in the widespread loss of promoter H3K27me3 and widespread corresponding increase in gene expression. However, only 137 genes exhibited an increase in gene expression using the isogenic p85b NLS mutant cell lines, which is**

inconsistent with the western data and suggests additional mechanism(s) may be play here. In order to rigorously test this hypothesis, H3K27me3 ChIP-seq should be performed to confirm that the differential gene expression is a result of genomic loci-specific changes to H3K27me3. Lastly, the logic as to why H3K27me3 was selected and followed up on is unclear, especially since AKT has already been shown to regulate H3K27me3 via EZH2 S21 phosphorylation (Cha, et al Science 2005).”

Response: Thanks for the insightful comments! We have now performed the suggested H3K27me3 ChIP-seq experiment as well as the RNA-seq experiment suggested below. The RNA-seq data shows that the expression levels of 3,224 genes are significantly up-regulated p85 β NLS mutant cells compared to the parental DLD1 cells. The H3K27me3 ChIP-seq revealed that the enrichment of H3K27me3 in 383 genes was reduced in p85 β NLS mutant cells compared to the parental DLD1 cells (Figure 5g). Given that the EZH containing PRC2 complexes are predominantly located in the heterochromatin regions. We also performed H3K27me3 ChIP-PCR on seven repeat sequences in different genomic regions in parental DLD1 and p85 β ^{KR-AA} mutant cells. Compared to parental DLD1 cells, the binding of H3K27me3 to four of the seven regions (D4Z4 repeat sequences on chromosome 4, satellite sequences on chromosome 1, Alu sequences on chromosome 19, and the telomeric TTAGGC repeats at the chromosome 7q) was drastically reduced, to two regions (satellite sequences on chromosome 4 and Alu sequences on chromosome 10) was reduced moderately, in p85 β NLS mutant cells (new Figure S5h). However, The binding of H3K27me3 to the promoter region of testis-specific histone 2B was not changed between parental DLD1 and p85 β NLS mutant cells (new Figure S5h). We postulate that the amount of p85 β binding to different chromatin regions varies, thereby modulating EZH and H3K27me3 differentially.

Regarding the logic of choosing H3K27me3, we postulated that the nuclear p85 β might regulate gene expression through certain epigenetic mechanisms. Thus, we screened the histone methylation sites, including H3K27, H3K4, H3K9, H3K36, and H3K76, in the parental and p85 β knockout cells and found only H3K27me3 levels were decreased in the p85 β knockout cells. We have now made this point clear in the revised manuscript. As to the report of regulation of H3K27me3/EZH2 by AKT, the Cha study showed that AKT phosphorylates EZH2 at the S21 site, thereby inhibiting its methyltransferase activity. This mechanism is distinct from the nuclear p85 β -USP7-EZH pathway we report here. Nonetheless, we have now discussed the Cha study in the revised manuscript.

4. “Similar to the above point, why does loss of p85b dramatically reduce both EZH1 and EZH2, but only influence the expression of 253 genes? EZH1/2 are the main H3K27 methyltransferases, with only the K3K9 methyltransferase G9a able to methylate H3K27 under select circumstances. Thus, one would hypothesize a greater impact on gene expression if EZH1/2 expression is destabilized.”

Response: As described above, our newly performed RNA-seq experiments showed that the p85 β NLS mutant significantly impacted the expression of 5268 genes rather than the 253 genes identified by the previous microarray experiment. It is well-documented RNA-seq is more sensitive than the microarray gene expression analysis.

5. “Linking to points 3-4 above, while it may be difficult to justify how such dramatic changes to EZH1/EZH2 expression and H3K27me3 results in limited changed to gene

expression, the manuscript could be strengthened by inclusion of the RNA-seq data in a main figure, and including the identity of the differentially regulated genes. This is especially true if key oncogenes/tumor suppressors are differentially regulated. This could be even further enhanced if the differentially regulated genes could be linked to changes to genomic loci-specific promoter H3K27me3.”

Response: We have now performed the suggested RNA-seq experiment and added the data in Figure 5a. Integrated analyses of RNA-seq and H3K27me3 ChIP-seq showed that the expression levels of 38 genes whose promoters reduced binding to an antibody against H3K27me3 in p85 β NLS mutant cells compared to the parental cells are up-regulated in p85 β NLS mutant cells (new Figure 5g). We validated the result by performing ChIP-PCR of H3K27me3 on the promoter region of DLG2 (Figure S5g), the top-ranked gene of the 38 genes.

6. “The cells utilized in Fig 5F appear to be wild type PIK3CA. However, as shown in Fig 3, only cells that express PIK3CA helical mutations are characterized by high p85b nuclear levels. Thus, if nuclear p85b is required for this mechanism described in this paper, it is essential that the cells utilized in experiments such as Fig 5F contain a helical PIk3CA mutation in the presence and absence of USP7 KO. Currently, the experiment shown in Fig 5F demonstrates that USP7 regulates EZH1/2 stability independently from PIK3CA helical domain mutant expression and may not fundamentally be different than what has been previously published (Lee J, et al, and Zheng, N, et al. Refs 32 and 33). This comment is also viable for Fig 5g and h.”

Response: The experiments described in the old Fig. 5f, g, and h were all performed in parental DLD1 cells, which harbor a PIK3CA E545K helical domain mutation. We have now made this point clear in the revised manuscript.

7. “The discussion does not include commentary on the existing literature that demonstrates EZH2 is an AKT substrate (Cha, et al Science 2005), and thus inhibition of EZH1/2 may support PI3K inhibition for more than just the proposed mechanism via USP7 stabilization.”

Response: We have included discussions of the aforementioned reference in the revised manuscript.

8. “Synergy calculations (Bliss/Loewe) could be performed to enhance the value of combination therapy in helical domain mutant cell lines and tumors, in comparison to PIK3CA wild-type and PIK3CA catalytic domain mutants.”

Response: We have now performed synergy calculations, and the data are shown in Figure S7h.

Minor points:

1. “Accessory experiments demonstrating the ability of E542/E545K to shift p85b into the nucleus could be performed with H1047R to solidify the proposed differences in molecular mechanisms by which PI3K mutations confer oncogenicity. E.g, overexpress H1047R in cells and evaluate p85b subcellular localization in a manner analogous to what was done for E545K in Fig S3E.

Response: The PIK3CA H1047R mutation was expressed and showed that expression of this kinase domain mutation did not shift p85 β into the nucleus (new Fig. S3f).

2. “Inclusion of IHC for subcellular localization of p85b in H1047R-mutated colon cancer specimens as part of Fig S3F would be beneficial”

Response: We have now performed the proposed experiment, and the data showed the p85 β is predominantly located in the cytoplasm of colon cancers with PIK3CA H1047R mutations (new Figure S3g).

3. “Why is it stated that global gene expression is being regulated if a total of 253 genes are differentially regulated in the p85bKR-AA cells?”

Response: As described in response to the major point 4 above, our newly performed RNA-seq experiment showed that the expression of 5268 genes was impacted.

4. “Text in the results section describing Fig 5A states, “... We thus examined histone modifications...” However, only H3K27me3 was evaluated in Fig 5. Fig 5 would benefit from inclusion of Fig S5b in the main figure 5.”

Response: We have now moved the old Fig. S5b into the main Figure 5C.

5. “As an extension of Figure 5, the conclusion would be strengthened by using DLD cells that express PIK3CA E545K with or without USP7 KO, and then evaluating whether H3K27me3 is preferentially reduced in the E545K + USP7 KO cells.”

Response: We have now performed the suggested experiment, and the data showed that knockout of USP7 in parental DLD1 cells with a PIK3CA E545K mutation reduced the levels of EZH1, EZH2, and H3K27me3 (new Figure 6 f).

6. “Tazemetostat is spelled incorrectly in the last sentence prior to the discussion.”

Response: This typo has been corrected.

7. “The very end of the discussion is redundant; describes the FDA approval of alpelisib and associated information twice in the same paragraph.”

Response: We have now deleted the redundant texts.

REVIEWER COMMENTS

Reviewer #1 (Remarks to the Author):

The revised manuscript contains a large amount of novel data that support the conclusions and address most of my concerns.

However, authors did not correctly interpret the first of my points.

I was intrigued by a function for p85 that emerges only in tumors with E545K PIK3CA mutation and not in healthy cells that underwent selective pressure and evolution. In the old and revised manuscript, authors implicitly claim that this function emerged only in tumors completely by chance. Nonetheless, having an NLS and being able to specifically bind a partner in the nucleus is hard to explain only by chance and with the exclusion of an evolutionary pressure that previously selected those sophisticated and diverse protein-protein interactions during evolution.

In my view, I would not exclude that p85 indeed functions alone in the nucleus of healthy cells of healthy individuals, with roles in epigenetic control shaped by evolutive pressure. Otherwise, why has the p85b NLS and p85b ability to recruit the deubiquitinase USP7 to stabilize EZH1/2 emerged and stabilized in the vertebrate lineage, from mice to humans?

I understand that this is subtle reasoning, but I think this might give a broader impact to the thesis claimed in the paper. I am happy to accept the manuscript if authors mention in the discussion the evolutionary problem of a complex function emerging new only in cancer and leave the door open to a potential role of nuclear p85 in healthy cells.

Reviewer #2 (Remarks to the Author):

The authors thoroughly address many reviewer comments and the manuscript is improved. For example, the inclusion of additional cell lines from different tumors and tissue of origin (PIK3CA helical mutants, PIK3CA catalytic mutants, and PIK3CA wild type) was critical, but the manuscript could still be improved by generating a PIK3CAH1047R mutation in the isogenic DLD cells. The logic for this lies in the heterogeneity associated with each of the cell lines selected, which is eliminated using the isogenic DLD line the authors have so elegantly constructed to evaluate PIK3CAE545K. The RNA-seq and H3K27me3 ChIP-seq experiment strengthened the manuscript but also created content that is not smoothly integrated with the main story (modulation of promoter H3K27me3 at the DLG2 tumor suppressor gene and a selection of other heterochromatic regions). Other improvements include subcellular fractionation experiments and IF using the NLS fused to GFP, and the more robust experiments evaluating EZH1/2 stability in Figure 5, all of which strengthen the authors' arguments.

Major points:

1. The manuscript may benefit from performing p85B NLS ChIP-seq and integrating with the RNA seq

and H3K27me3 ChIP-seq.

2. It is unclear why the isogenic DLD PIK3CAH1047R cells were not engineered, which would provide the most linear comparison between helical and catalytic domain PIK3CA mutants? They could have been used instead of, or in addition to, the additional cell lines utilized in Figures 2, 3, etc.

Minor:

1. Inclusion of the additional cell lines in figure 2 is nice. To clarify for readers, more clearly labeling figures 2f, g, h, and I to indicate which mutation these cells contain. E.g., "H240 PIK3CAE545K."

2. Line 340 – typo "PRC2 complex"

3. The paragraph added in the results section (starting with line 344) referring to H3K27me3 ChIP-seq data is dense and could benefit from streamlining for clarity.

** See Nature Research's author and referees' website at www.nature.com/authors for information about policies, services and author benefits.

Responses to the reviewers' comments

We thank the reviewers for their high enthusiasm for our manuscript and for the acknowledgment that the revised manuscript was much improved with a large amount of new data. We also appreciate their additional comments, which help to improve the manuscript further. New additions/changes are highlighted in red in the revised manuscript. Below is our point-by-point response to their concerns:

Response to comments of Reviewer #1

“The revised manuscript contains a large amount of novel data that support the conclusions and address most of my concerns.

However, authors did not correctly interpret the first of my points.

I was intrigued by a function for p85 that emerges only in tumors with E545K PIK3CA mutation and not in healthy cells that underwent selective pressure and evolution. In the old and revised manuscript, authors implicitly claim that this function emerged only in tumors completely by chance. Nonetheless, having an NLS and being able to specifically bind a partner in the nucleus is hard to explain only by chance and with the exclusion of an evolutionary pressure that previously selected those sophisticated and diverse protein-protein interactions during evolution.

In my view, I would not exclude that p85 indeed functions alone in the nucleus of healthy cells of healthy individuals, with roles in epigenetic control shaped by evolutive pressure. Otherwise, why has the p85b NLS and p85b ability to recruit the deubiquitinase USP7 to stabilize EZH1/2 emerged and stabilized in the vertebrate lineage, from mice to humans?

I understand that this is subtle reasoning, but I think this might give a broader impact to the thesis claimed in the paper. I am happy to accept the manuscript if authors mention in the discussion the evolutionary problem of a complex function emerging new only in cancer and leave the door open to a potential role of nuclear p85 in healthy cells.”

Response: Sorry for misinterpreting your prior comments. We took your suggestion and added the following sentences into the discussion section: “Although our data suggest that the nuclear p85 β -USP7-EZH1/2 pathway is activated in cancer cells with a PIK3CA helical domain mutation, we cannot rule out the possibility that this pathway could also be triggered by certain physiological stimuli in normal cells as well.”

Response to comments of Reviewer #2

Major points:

1. “The manuscript may benefit from performing p85 β NLS CHIP-seq and integrating with the RNA

seq and H3K27me3 ChIP-seq.”

Response: We, too, wanted to perform a p85 β ChIP-seq experiment. We actually attempted to do it when we performed the H3K27me3 ChIP-seq. However, the quality control analyses showed the p85 β -specific antibody failed to immunoprecipitate chromatin DNA efficiently (Figure 1 below), thereby preventing us from performing the experiment. It is worth noting that we screened several commercially available anti-p85 β antibodies and were able to validate only this one with the p85 β knockout cell lines that we generated. We postulate that this p85 β -specific antibody is not suitable for ChIP-seq for some technical reasons. Alternatively, the binding p85 β to chromatin is of low affinity or indirect.

2. “It is unclear why the isogenic DLD PIK3CAH1047R cells were not engineered, which would provide the most linear comparison between helical and catalytic domain PIK3CA mutants? They could have been used instead of, or in addition to, the additional cell lines utilized in Figures 2, 3, etc.”

Response: We realize that we might not clearly explain the isogenic DLD1 cell lines. The parental DLD1 cell line has a heterozygous PIK3CA E545K mutation (labeled as parental in Figure 1a). The isogenic DLD1 PIK3CA WT-only cell line (Figure. 1a) was generated by knocking out the mutant allele, whereas the DLD1 PIK3CA mutant-only cell line (Figure 1a) was generated by knocking out the WT allele. In this context, knocking in a PIK3CA H1047R mutation in DLD1

cells would be ectopic. Moreover, we have provided abundant evidence in the manuscript that **in cancer cell lines with a PIK3CA H1047R kinase domain mutation: 1) p85 β does not dissociate from PI3K complexes (Figure 1 f to h, and Figure S1i); 2) knockout of p85 β did not impact tumor growth (Figure 2 g to h); 3) p85 β did not translocate into the nucleus to stabilize EZH1 and EZH2 (Figure 3 f & g, Figure 5e); and 4) EZH inhibitor tazemetostat did not significantly inhibit tumor growth (Figure 7e).**

Minor:

1. “Inclusion of the additional cell lines in figure 2 is nice. To clarify for readers, more clearly labeling figures 2f, g, h, and I to indicate which mutation these cells contain. E.g., H240 PIK3CAE545K.”

Response: We have now labeled the cell lines as suggested.

2. Line 340 – typo “PRC2 complex”

Response: Sorry for the typo. We have now corrected

3. “The paragraph added in the results section (starting with line 344) referring to H3K27me3 ChIP-seq data is dense and could benefit from streamlining for clarity.”

Response: We now rewrote this section and made it a new section under its own subheading. I hope these revisions make it easier to read.

REVIEWERS' COMMENTS

** See Nature Portfolio's author and referees' website at www.nature.com/authors for information about policies, services and author benefits